# Copy number architectures define treatment-mediated selection of lethal prostate cancer clones

A. M. Mahedi Hasan [1,41], Paolo Cremaschi [1,41], Daniel Wetterskog [1,41], Anuradha Jayaram[1,2], Stephen Q. Wong[3,4], Scott Williams[4], Anupama Pasam[3], Anna Trigos [3], Blanca Trujillo [1,2], Emily Grist[1], Stefanie Friedrich[1], Osvaldas Vainauskas[1], Marina Parry [1], Mazlina Ismail [1], Wout Devlies [1], Anna Wingate[1], Mark Linch[1,2], Cristina Naceur-Lombardelli [1], PEACE consortium*, Charles Swanton [5,6,7], Mariam Jamal-Hanjani [5,7,8], Stefano Lise[1], Shahneen Sandhu[3] & Gerhardt Attard [1,2] ✉

Despite initial responses to hormone treatment, metastatic prostate cancer invariably evolves to a lethal state. To characterize the intra-patient evolutionary relationships of metastases that evade treatment, we perform genome-wide copy number profiling and bespoke approaches targeting the androgen receptor (AR) on 167 metastatic regions from 11 organs harvested post-mortem from 10 men who died from prostate cancer. We identify diverse and patient-unique alterations clustering around the *AR* in metastases from every patient with evidence of independent acquisition of related genomic changes within an individual and, in some patients, the co-existence of *AR*-neutral clones. Using the genomic boundaries of pan-autosome copy number changes, we confirm a common clone of origin across metastases and diagnostic biopsies, and identified in individual patients, clusters of metastases occupied by dominant clones with diverged autosomal copy number alterations. These autosome-defined clusters are characterized by cluster-specific *AR* gene architectures, and in two index cases are topologically more congruent than by chance (*p*-values $3.07 \times 10^{-8}$ and $6.4 \times 10^{-4}$). Integration with anatomical sites suggests patterns of spread and points of genomic divergence. Here, we show that copy number boundaries identify treatment-selected clones with putatively distinct lethal trajectories.

Deciphering the conundrum of treatment resistance in metastatic epithelial malignancies is a major unmet medical need. Prostate cancer shows high response rates to androgen deprivation therapy (ADT), but relapses often occur after an average of 2 years[1]. Selection of clones harboring *AR* amplification or mutations, structural rearrangements, splice variants and a plethora of events consistent with treatment-mediated selection to maintain AR activity, despite medical efforts to inhibit it, result in an often rapidly lethal state that remains poorly understood[2–5]. *AR* alterations in liquid or tissue biopsies associate with shorter responses to second-line second-generation hormonal treatments[6,7] and new drugs are in development to target aberrant AR[8–10]. However, heterogeneity of *AR* alterations across metastases could create a challenge that complicates their utility as a biomarker or therapeutic target.

A full list of affiliations appears at the end of the paper. *A list of authors and their affiliations appears at the end of the paper. ✉e-mail: g.attard@ucl.ac.uk

                                                                 

In this work, we characterize *AR* genomic complexity across spatially-separated lethal metastases, and by using pan-autosome copy number features, evaluate the relationships of intra-patient metastases to inform on their evolutionary and metastatic trajectories.

## Results

### High selective pressure for structural alterations involving the *AR* gene and its enhancer following treatment with abiraterone or enzalutamide

We performed rapid post-mortems on 10 men who died from metastatic castration-resistant prostate cancer (mCRPC). Over the course of their treatment, between 2001 and 2019, all men had developed treatment resistance, defined by a rise in serum prostate specific antigen (PSA) on second-generation AR signaling inhibitors (abiraterone or enzalutamide, Fig. 1a, Supplementary Data 1). In total we harvested 201 fresh frozen tumor samples from 11 different organs. We also collected plasma from nine men at death and retrieved 33 archived formalin-fixed paraffin embedded (FFPE) tumor samples from eight of the men. These FFPE tumor samples were obtained at diagnosis or after development of castration-resistance, either for palliation or as a requirement for participating in a clinical trial.

Copy number alterations are common in advanced prostate cancer[11]. We first used shallow whole-genome sequencing (median coverage: 0.34X, range: 0.03X-5.88X, Supplementary Data 2) to estimate tumor fraction and obtain copy number profiles (see Supplementary Fig. 1 for experimental study design). For tumor fraction estimation, we used the consensus of two approaches: a previously published method implemented in the ACE package[12] and, secondly, a bespoke strategy that derived a measure of tumor content based on the allelic imbalance at heterozygous single nucleotide polymorphisms (SNPs). To maximize the accuracy of the latter approach, we concurrently performed whole-genome sequencing (median coverage: 24.28X, range: 14.14X-32.62X) on the 10 patients' germline samples (described in methods). These methods led us to exclude samples with a tumor fraction of <0.2 (not including), namely 34 (17%) fresh-frozen post-mortem, four plasma (44%) and nine formalin-fixed archival (27%) samples. Of the resultant formalin-fixed samples, for six patients we had standard-of-care diagnostic prostate biopsies, for two patients we had only CRPC biopsies (brain and prostate) and for one patient we had a diagnostic biopsy, a prostatectomy sample obtained 18 months after the first biopsy sample was taken and a mCRPC liver biopsy six months before death (Fig. 1a).

We then extracted *AR* copy number from the remaining 167 post-mortem samples. Of these, 74 (44%) had gain of the *AR* (defined as copy number ≥2 and involving <80% of Xq, also see Methods, Fig. 1b), confirmed by using droplet digital PCR (ddPCR) or targeted NGS (Supplementary Fig. 2), including at least one metastasis from eight out of 10 men. We observed distinct groups of metastases with overlapping patterns of chromosome X copy number architecture showing intra- and inter-patient diversity (Supplementary Fig. 3). We also noted that variably large areas of chromosome X showed copy number gain (Fig. 2). Invariably when copy number change occurred, it involved the *AR* and in metastases from CA34, CA43, CA63, CA79 and PEA172, also its associated (centromeric) enhancer, which in its gained state, was recently shown to be associated with resistance to abiraterone or enzalutamide[13,14].

We then used the above information to select two to six metastases from each patient for resequencing at a higher depth of coverage (total: 25 samples; median coverage: ~60X; range: 27X – 83X, Supplementary Data 2). Focusing on chromosome X in these data, we identified breakpoints that were unique to each patient and occurred at a high density within a large genomic area (30 Mbp, henceforth referred to as "*AR* locus") around *AR* (Fig. 3a and Supplementary Fig. 4). This inter-patient diversity is of relevance to efforts of investigating *AR*-associated breakpoints as biomarkers for tracking tumor clones and

treatment selection that will require patient-specific probes or broad approaches to sequence chromosome X at sufficient depth[3,15]. In the first instance, we noted overlapping break-point occurrences in spatially separated metastases (for example, CA34 right and left liver lobes; and CA63 vertebra and soft tissue of rib) which clearly showed that the same clone spread to different sites. Intriguingly, we found that patients (CA27, CA36 and CA83) with the shortest exposure (2, 3, and 2 months respectively compared to 4, 6, 10, 11, 12, 22 and 32 months for the rest of the cohort) to first-line second-generation hormonal treatment (abiraterone or enzalutamide, Supplementary Data 1) had the fewest break-points detected at the *AR* locus (median of 2 versus 10 unique break-points per patient respectively, *p*-value 0.009). We also observed intra-patient breakpoint heterogeneity in both the same and different organs, including different combinations of unique breakpoints in two adjacent samples from a tumor at the bladder base in CA63 and in a thoracic lymph node and dura from CA79. This intra-patient break-point diversity signifies convergent evolution for alterations involving the *AR* region following treatment selection pressures exerted by hormone therapies.

To further study intra-patient breakpoint diversity, we designed patient-specific custom NGS probes (Supplementary Data 10) and performed high-coverage NGS on samples from patient CA34, who had the highest density of breakpoints converging on *AR* (Fig. 3b). We included probes for the break-point associated with a *TMPRSS2-ERG* fusion that we confirmed was present in all tumors analyzed (putatively an early event that preceded acquisition of structural rearrangements converging around *AR*). The clone most abundant in liver metastases and its "gatekeeper" lymph node harbored at least six *AR* break-points unique to this patient (BP1 to BP6), admixed at varying but considerably high proportions. We also identified evidence of subclones in two external lymph node metastases (CA34_3, CA34_2) and a prostate biopsy (CA34_1) with varying but notably lower proportions of BP1-BP6.

There was no evidence of a focal gain at the *AR* locus in CA27 metastases. However, we detected an inversion involving exon 5 to 7 of *AR* that was previously shown to result in an *AR* splice variant that lacked ligand-binding capability and was constitutively activated independently of ligand[16] (Fig. 3c). Using customized high-coverage targeted NGS and ddPCR, we confirmed the copy-number neutral breakpoint to be present in every metastasis and in one of the four prostate tumor regions at varying fractions (at sub-clonal level) admixed with *AR* neutral cells (Fig. 3c). Overall, we effectively confirmed diverse and patient-unique alterations clustering around the *AR* and its enhancer in metastases from every patient, supporting the potent selective pressures for *AR* aberrant clones in men receiving hormonal therapies.

### Convergent evolution of independent *AR* somatic point mutations

Hormone treatment pressures can also select clones harboring *AR* somatic point mutations[5]. We performed ~1400X targeted NGS designed to capture *AR* coding sequences and recurrent prostate cancer mutations and in total detected seven unique non-synonymous functionally-active mutations within *AR* coding regions. Of these, six (encoding for T878A, C687Y, H875Y, D891N, G751C, and V716M) occurred in CA36 and one (E710G) in CA43. In CA43, we observed gain of the mutant pE710G allele in four of six and of the wild-type allele in two of six metastatic samples (Supplementary Data 3). In CA36, (one of 10 patients, 10%), pT878A was detected in every metastasis (Fig. 3d). Prior studies suggest this mutation is the most common in mCRPC patients, detected in 10–15% of the cases[7]. We detected all *AR* somatic mutations in plasma that were harbored in extra-cranial metastases but not pC687Y which was detected solely in the brain metastases.

We hypothesized that all metastases could either have been seeded by the same clone harboring a T878A mutation or the clones

                                                                                      

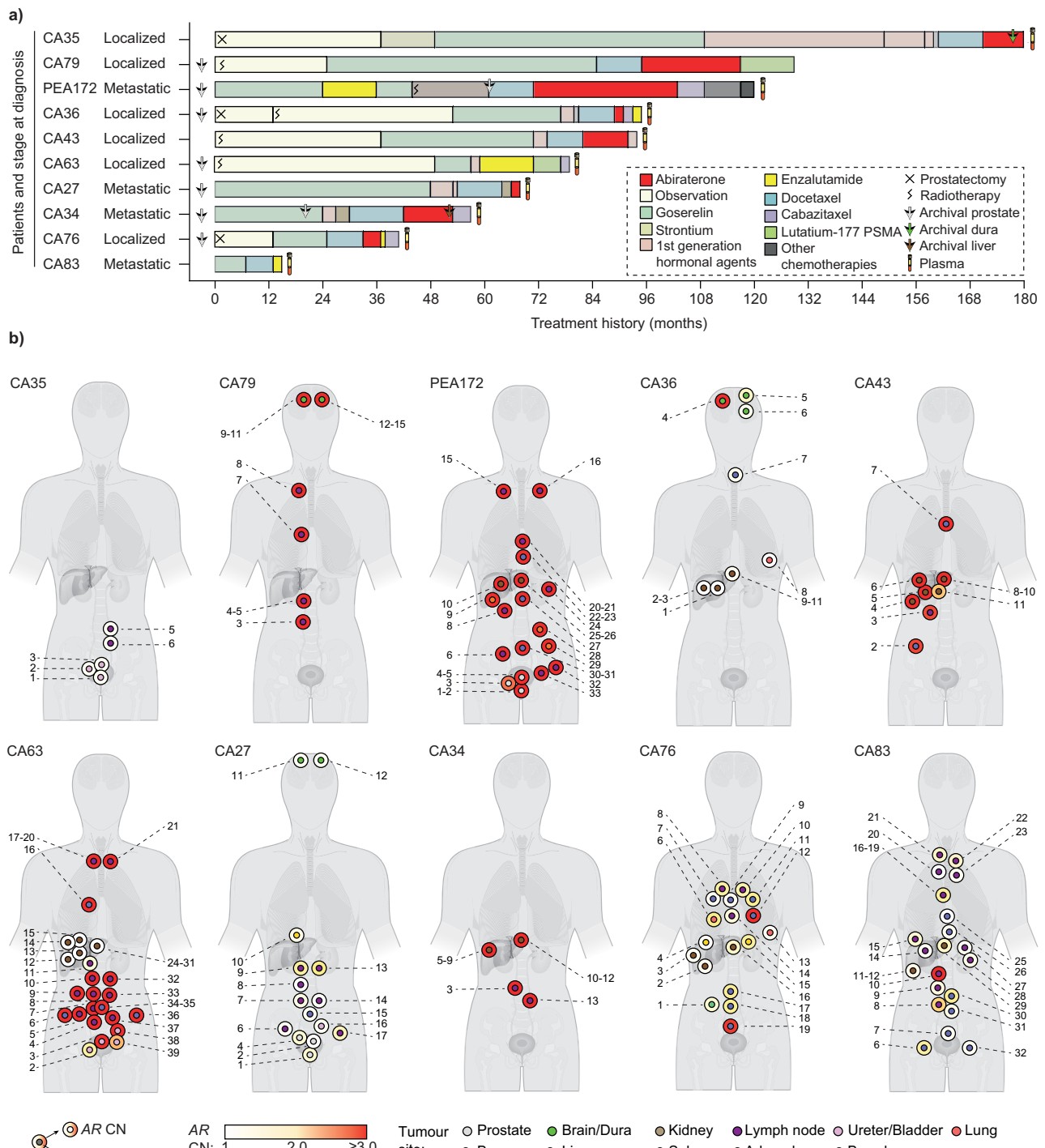

**Fig. 1 | Treatment regimen, anatomical positions and *AR* copy number status of metastases harvested post-mortem. a** Horizontal stacked bars denote the treatment regimen administered, color codes in the legend. Patients are arranged based on the time from diagnosis to death. Arrows denote the time of tissue biopsy (prostate, brain or liver) at initial diagnosis or as part of standard-of-care. Plasma was collected post-mortem from nine patients, denoted by blood collection tubes. **b** *AR* copy number state and approximate anatomical positions of tumors with tumor content ≥0.2. Numbers along a human body denote the identity of patient-wise tumors harvested post-mortem.

independently acquired it after metastatic spread. In our search for evidence supporting either hypothesis, we identified in one of the liver metastases (CA36_11) alleles with mutation(s) encoding for D891N or T878A or both (Fig. 3e). Specifically we identified 364 sequencing reads harboring mutations at both positions X:66943591 (that codes for p.D891N) and X:66943552 (coding for p.T878A) in the same sample with reads that were mutant at one of these positions but wild-type at the other (6 reads with wild-type p.D891 but mutant p.T878A, and 393

reads with mutant p.D891N but wild-type p.T878, Fig. 3e). Although this means that cells with one mutation acquired the other, we cannot distinguish the order of events. To further investigate the biological pressures for the selection of these two AR mutations, we used a reporter luciferase construct in prostate cancer cells transfected with AR wild-type or a combination of the detected mutations and treated with a range of ligands, including progesterone and prednisone previously suggested as contributing to resistance against abiraterone[17–19]

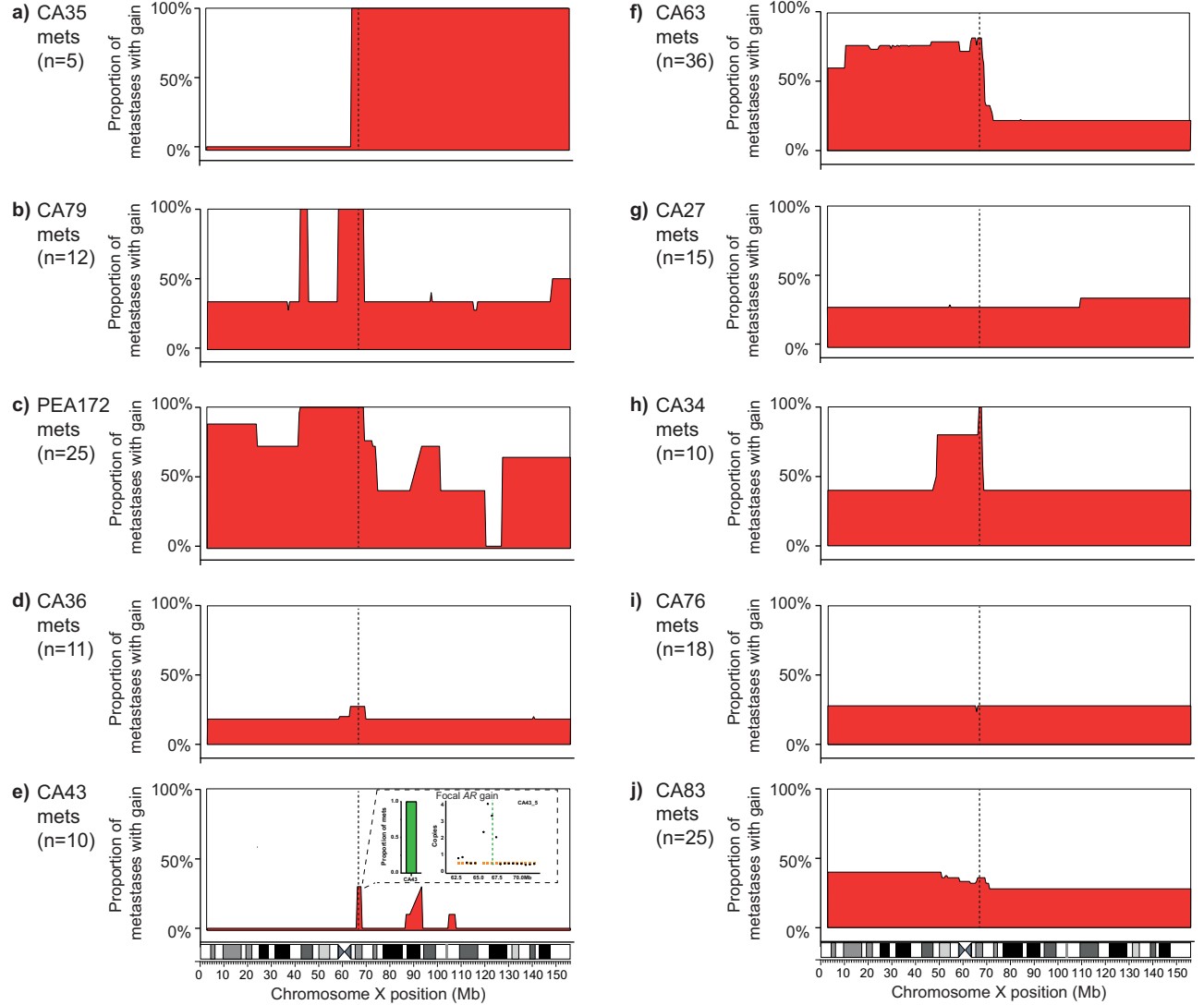

**Fig. 2 | Chromosome X architecture captures intra- and inter-patient diversity.**
**a–j** Skyline plots (drawn using Graphpad prism) showing the proportion of samples in each patient with copy number gains (≥2 copies, along y-axis) at individual bins (500 kb wide) across chromosome X (along *x*-axis) (Supplementary Data 6). Number of metastatic samples per patient (*n*) is provided for each panel. Vertical dotted lines denote position of *AR* gene and associated enhancer region. **e** Inset used to show focal gain in *AR*, not evident at low resolution in the skyline plot. Orange squares depict mean copy number for contributing segment(s) and black dots represent copy number of each bin. Number of metastatic samples used per patient - CA35: *n* = 5, CA79: *n* = 12, PEA172: *n* = 25, CA36: *n* = 11, CA43: *n* = 10, CA63: *n* = 36, CA27: *n* = 15, CA34: *n* = 10, CA76: *n* = 18, CA83: *n* = 25.

(Fig. 3f). We found that acquiring AR p.D891N alone did not confer significant increased activation by a range of ligands compared to AR wildtype. Overall, this suggests that despite being present in all metastases analyzed, the p.T878A mutation occurred after metastatic seeding and in this liver metastasis emerged independently in a lineage with p.D891N mutation. Based on our functional assays, it may be more likely that p.D891N mutant cells acquired a p.T878A than vice versa. We deemed the alternative explanation of reverting a mutation to its wild-type as unlikely although we are unable to exclude this possibility. We also observed evidence of a second independent event in the right dural metastasis (CA36_4) with an amplification of the T878A *AR* mutant allele (Fig. 3d).

If the acquisition of an *AR* mutation occurred at random across metastases from any patient, we would have expected a uniform distribution of *AR* mutations across all metastases in our cohort. In contrast, we observed that all metastases analyzed from CA36 harbored at least one functionally-relevant *AR* mutation that we showed was acquired via independent events (Fig. 3e), whilst the majority of patients (8 out of 10) did not have a mutation of interest in any

metastasis. This introduces a hypothesis that a sub-set of prostate cancers (equivalent to ~15%) have an evolutionary course that converges on an *AR*-mutant resistant genotype following hormone therapy.

## Autosomal copy number transition points define the relationships of individual metastases and identify clonally-related clusters

To interrogate the evolutionary paths followed by groups of metastases with different *AR* gene architectures we deliberately focused on copy number change that occurred in the autosome. In several cancers, parts of the genome are affected by copy number alterations that can show an evolutionary order to their occurrence across patients and cancer types[20]. Although involved regions are repeated across cancers, the starting event that leads to each change could be different in every patient's tumor. We posited that the junction in the genome where copy number change occurred could therefore offer an opportunity for "tumor fingerprinting" to track relationships over time and space. Using copy number profiles derived from whole-genome

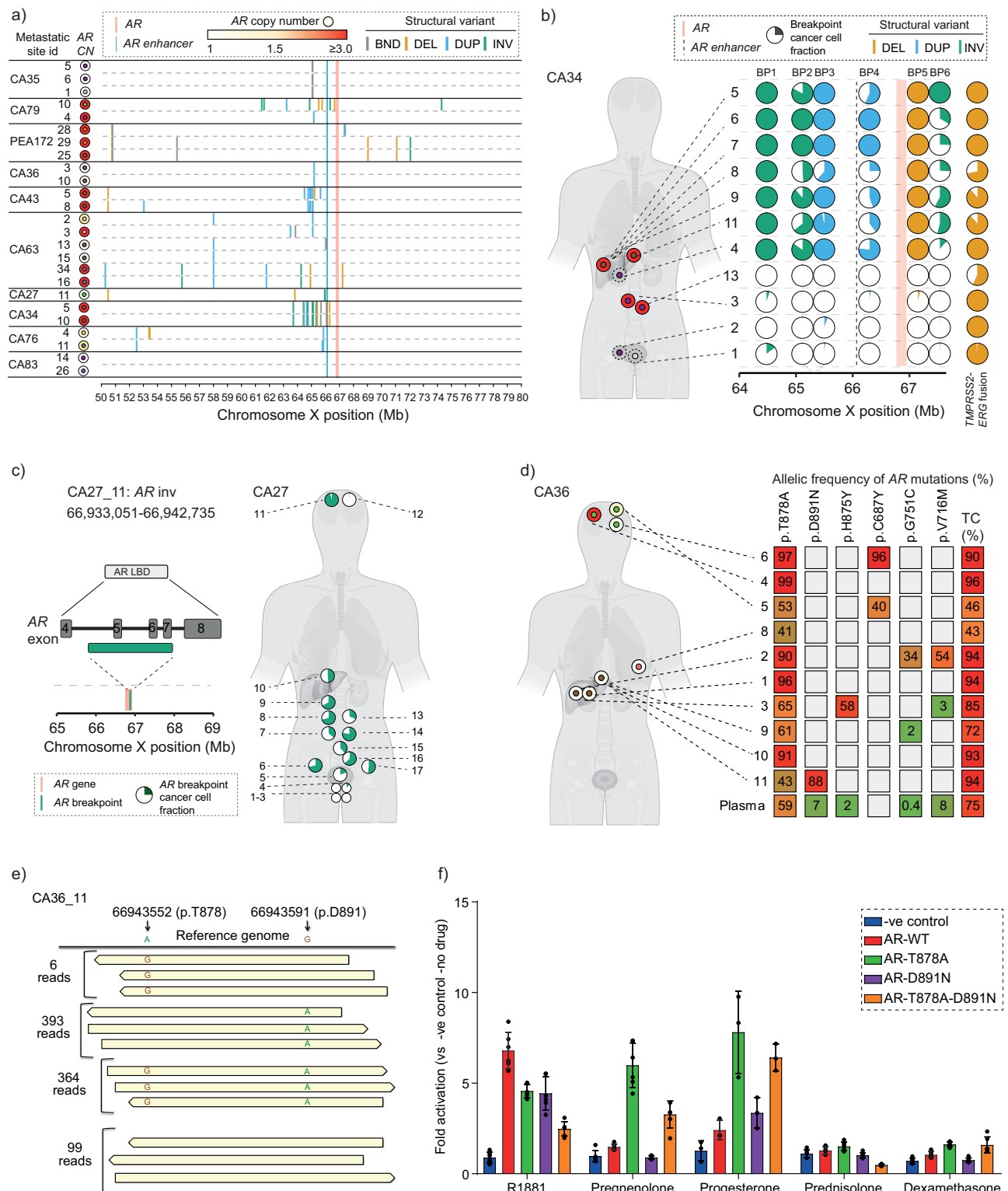

sequencing, we identified the boundaries at which a change in the copy number of adjacent autosome segments occurred (henceforth referred to as "transition points").

We investigated the co-occurrence of transition points in temporally-separated archival FFPE biopsies (24 from eight patients, Supplementary Data 2) with tumors harvested post-mortem. Whilst we observed a limited sharing of individual transition points across metastases from different individuals, we uniquely observed that transition points in archival FFPE samples from an individual were shared with all metastases harvested post-mortem from the same patient (putatively patient-common or truncal transition points) (Fig. 4a). This reaffirms a common clone of origin in metastases at death[21] and, now shown here, the same clone of origin detected in diagnostic biopsies (Supplementary Fig. 5).

When we compared plasma at death and metastases obtained post-mortem for five patients, we identified that in four out of five cases (CA34, CA36, CA76 and CA83) > 90% of transition points in plasma were detected in the metastases analyzed, suggesting that

**Fig. 3 | High selective pressure for genomic alterations involving the *AR* gene region. a** Breakpoints at the start of structural variants (BND: translocation, DEL deletion, DUP duplication/gain and INV inversion) detected in high-coverage (~60X) whole-genome next-generation sequenced 25 samples from ten individuals, showing a 30 Megabase region of chromosome X around *AR* gene (Supplementary Data 7). **b** Left panel: anatomical positions of metastatic samples (inner circle) and their *AR* copy number (outer circle) are shown for CA34. Dashed outer circles depict samples with tumor content <0.2. Right panel: pie charts depicting the clonality (by cancer cell fraction, CCF) of breakpoints in the proximity of the *AR* gene and on chromosome 21 at the position of *TMPRSS2:ERG* fusion detected by high-coverage targeted sequencing on 11 samples (Supplementary Data 12). *AR* and associated enhancer are depicted with vertical salmon and blue dashed lines, respectively. We do not have high confidence in cancer cell fraction (CCF) calling for samples (CA34_1, CA34_2 and CA34_3) with very low tumor content. **c** Left panel: High-coverage custom NGS confirmed a copy number neutral breakpoint (for an inversion event) involving exon 5 to 7 of *AR* gene in patient CA27 which resulted in a ligand-independent, constitutively activated AR splice variant. Right panel: the distribution of the sub-clonal breakpoint and associated CCF at different anatomical sites are shown in pie charts (Supplementary Data 7). **d** Left panel: Anatomical sites of tissue sampled (color coded inner circles as for Fig. 1b) are shown with *AR* copy numbers (outer circles). Right panel: Pathological mutations detected in CA36 metastases using high-coverage targeted sequencing. Allelic fractions and tumor content are indicated by color and number, respectively. **e** Independent acquisition of *AR* mutations in a liver metastasis (CA36_11) detected using amplicon-based, high-coverage targeted sequencing. The supporting reads confirming each allele type shown on left: wild-type alleles are shown on top and mutated alleles are shown along the cartoon reads. **f** Reporter-luciferase assay showing activation of wild-type and mutant AR (T878A and D891N, individually and combined) by clinically relevant ligands (R1881: a synthetic androgen). Biological replicates of reporter-luciferase assay have been provided in Supplementary Fig. 9.

metastatic sampling captured the majority of plasma DNA shedding clones (Fig. 4b). However, in PEA172, ~43% of transition points in plasma were not detected in any of the 25 metastases studied.

We then compared transition points in CA27 metastases and the prostate tumors harvested at death split by whether the *AR* inversion break-point was detected (Fig. 3c): although all tumors shared the majority of transition points, the two break-point negative prostate tumors had a high proportion of transition points that were not shared with any metastases in contrast to the break-point positive prostate tumor (and all other metastases) (Fig. 4c). This, along with the finding from targeted sequencing of *AR* breakpoints (Fig. 3b) suggests that prostate tumors harvested at death were clonally different from other tumors present in the same patient. Although this may be explained by evolution of primary tumors after seeding of metastases[22], it may also support a hypothesis that the prostate microenvironment can select for or allow the survival of different clone(s) compared to metastatic sites.

In addition to truncal transition points shared across metastases from the same patient, in some patients we noted transition points shared across distinct groups (or clusters) of metastases. To use these data to determine the evolutionary relationship of metastases, we developed a hierarchical clustering-based algorithm (we refer to as Start of Copy number change for Relationship Assessment and Testing Clone Histories, SCRATCH) that determined the relationship of temporally and spatially-separated tumors (hence named "SCRATCH relational network") based on the correlation of the copy number values at transition points (Fig. 4d, Supplementary Data 4, Supplementary Fig. 6). This framework attempted to split metastases into groups that maximized the inter-cluster and minimized the intra-cluster distances based on the copy number differences at the transition points (see Methods). We observed that tumors in all patients were assigned to a limited number of clusters (two or three), regardless of the number of tumors analyzed, including for example CA63 where 36 autopsy samples formed three (Fig. 4d) and in PEA172 where 25 autopsy samples formed two clusters (Supplementary Fig. 6). We propose each cluster had a distinct dominant clone separating them from other clusters, which represents evolutionary genomic divergence.

### Dominant autosomal copy number changes define a limited number of clusters of metastases that share clonal mutations

In a patient, if metastases in the same SCRATCH-defined cluster had evolved separately from those in a different cluster, we expected that they would share the same clonal mutations. To test this, we first confirmed metastatic samples were assigned to the same cluster regardless of sequencing depth (Supplementary Fig. 6). We then extended our hypothesis that the correlation distance (or node distance) between a pair of samples in a patient based on transition points would correlate with the overlap in non-silent clonal mutations arising from the dominant clone. We applied the SCRATCH algorithm and extracted the node distance for metastases with high-coverage WGS and using a previously described tool, Sclust[23], that controls for tumor fraction and ploidy, defined clonal mutations in each patient (Supplementary Data 5). We then calculated the intersection of total non-silent clonal mutations in pairs of metastases ($N = 27$ pairs) from the same patient, controlled for the total number of the smaller set of mutations between the compared metastases, and confirmed a difference in the number of clonal mutations in metastases in the same SCRATCH-defined cluster from those in other clusters (p-value 0.0012) (Fig. 4e, f). This suggests that deriving tumor clone relationships using copy numbers at transition points is consistent with assumptions made using clonal mutation analysis.

### *AR* structural alterations occur in established clones with distinct autosome copy number profiles

Given *AR* alterations are selected by hormone treatment, we posited that these are relatively late events in prostate cancer evolution, emerging from established clones at the development of resistance to ADT. To investigate this, we used congruence models to test whether a cluster of metastases derived from autosome transition points was more likely to have the same *AR* structural alterations. This required cases with sufficient metastases namely, CA63 and PEA172. By separately using both the Baker's Gamma index[24] and Congruence Index ($I_{cong}$)[25] on patients CA63 and PEA172, we observed that clusters of metastases defined by their autosome were indeed associated with metastases in the same cluster defined by their chromosome X copy number profile, irrespective of anatomic sites (Fig. 5a, b). These relationships were topologically more congruent than by chance (p-value $3.07 \times 10^{-8}$ and $6.4 \times 10^{-4}$ for CA63 and PEA172, respectively). This was also confirmed by an informal annotation that metastases in the same cluster had the same *AR* copy number status. This observation supports a model of expansion of a limited number of distinct clones each defined by common transition points that acquire unique *AR* alterations at the development of resistance to therapy.

### Metastatic trajectories to lethal disease

We compared AR expression data from RNA sequencing on metastases showing intra-patient differences in *AR* status ($N = 39$ from CA63, CA76, CA83 and PEA172) with AR transcriptional activity (AR score) (Fig. 5c). We found a weak but positive correlation between *AR* copy number gain and normalized AR expression (p-value 0.009) and between *AR* copy number gain and AR score (p-value 0.03, Supplementary Fig. 7) but when analyzing tumors grouped by patient, we observed a more complex picture. There was no difference in AR score across tumors with low and high levels of AR expression in three patients (CA63, CA76 and CA83) whilst in PEA172, characterized by high *AR* copy numbers and consequently relatively high AR expression in all tumors, AR expression and score were correlated (p-value 0.003)

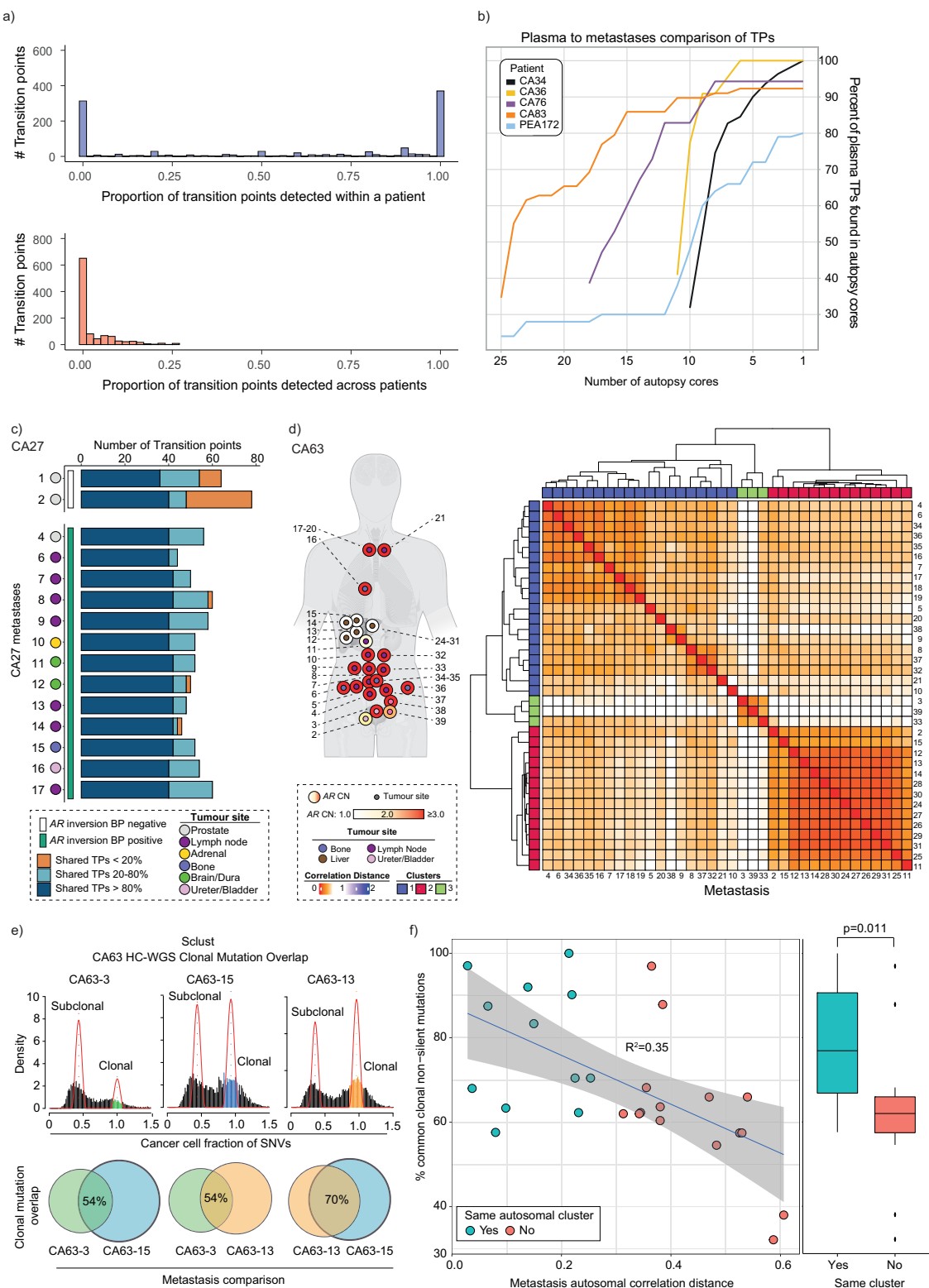

(Fig. 5c, d and Supplementary Fig. 8). We confirmed consistently high expression levels of the AR-response genes *KLK3* and *TMPRSS2* in tumors from CA63, CA76 and CA83 with and without *AR* gain (and respectively high and low AR expression) using ddPCR (Fig. 5d and Supplementary Fig. 8). We also found in PEA127, in keeping with high *AR* copy number levels, high AR, KLK3 and TMPRSS2 expression across all tumors, with higher KLK3 levels in cluster 1 versus cluster 2, confirming differential AR activity between metastases from the two *AR*-gained but structurally distinct clusters.

In CA63, of the two biopsies from the same bladder tumor, one (CA63_3) was *AR* amplified and the other (CA63_2) was not. Clustering based on their autosome copy number transition points assigned the two adjacent tumors to separate clusters with *AR* copy number matching the respective bladder tumor samples. This suggested that at a point in the evolutionary timescale of cancer progression, the CA63_3 clone acquired a genomic event that amplified *AR* and at death was the most abundant clone in the bone and lymph node metastases whilst the liver metastases and porta hepatic lymph nodes were populated by

**Fig. 4 | Copy number transition points confirm the same origin of lethal prostate cancer and archival biopsies and define the relationships of lethal metastases. a** Histograms showing sharing of copy number transition points detected in archival biopsies (total 24 archival biopsy samples from 8 patients) with metastases harvested post-mortem from the same patient (upper panel) and for each archival sample, across metastases harvested from each of the remaining patients (lower panel). A bin-width of 0.02 is chosen for both histograms along $x$-axis. **b** Percent of transition points detected in plasma (tumor content $\geq 0.2$), collected post-mortem, are plotted as a function of the number of post-mortem metastatic samples from the same patient ($N = 5$ patients) (Supplementary Data 8). **c** Stacked bars show different percentages (<20%, 20 to 80% and >80%) of shared copy number transition points among autopsy samples in CA27 (Supplementary Data 8). Two prostate tumors (1 & 2) do not share a pathogenic *AR* inversion break point and display a higher percentage of tumor-unique transition points, while the remaining prostate tumor (CA27_4) shows more homogeneity of shared transition points with other distal metastases and, in unison, share the break point. **d** Left panel: Metastatic samples harvested postmortem in CA63 are depicted with

anatomical position (inner circle) and *AR* copy number (outer circle). Right panel: Post-mortem metastatic samples form three distinct clusters by applying the SCRATCH clustering algorithm (described in methods). The color scheme of the heatmap is based on the correlation distances calculated using copy numbers at transition points. **e** Illustrative figure showing how clonal mutations (Supplementary Data 5), detected using Sclust, form a distinct peak at cancer cell fraction of 1.0. Intersection of non-silent clonal mutations between two samples were chosen and normalized by the total number of the smaller set of such mutations (described in methods). **f** Left panel: Correlation of percent common clonal non-silent mutation between metastases pairs in comparison ($N = 27$ comparing pairs) from the same patient and corresponding metastasis autosomal distances are shown with linear trendline (shaded area represents the 95% confidence interval level). Right panel: Box plot showing the distribution of percent common clonal non-silent mutations ($N = 27$ comparing pairs) by assignment of post-mortem samples in a cluster. Whisker follows mean $\pm$ IQR * 1.5 format of each box. Willcoxon non-parametric (one-sided) test was used to measure significance of difference between those two distinct groups, as shown.

the non-amplified clone shared with CA63_2 (Fig. 5e). The anatomical proximity of the bladder to the prostate and the retained admixture of the two distinct clones in opposite sides of the same tumor lesion could suggest that at least two of the three dominant clones at death in anatomically-distinct metastases originated independently from this site. The patient underwent a period of 48 months observation after surgery, prior to start of ADT. We detected a large number of transition points shared in either bladder area, and all metastases in the same respective cluster suggesting continued copy number evolution prior to spread that we posit occurred in minimal residual disease in the bladder wall that remained after surgery. There was a notable difference in AR expression in gained compared to non-gained metastases with the exception of the non-gained bladder tumor (adjacent to an area with gain suggesting the possibility of admixture of *AR* gained and non-gained clones) (Fig. 5d).

We observed a similar pattern of divergence with two dominant clusters of metastases in patient PEA172. Two anatomically adjacent metastases in the bowel (PEA172_28 and PEA172_29) had distinct *AR* architecture and based on their autosome clustered separately with other metastases of similar *AR* architecture. Tumor PEA172_28 clustered predominantly with lymph nodes while tumor PEA172_29 clustered with liver metastases. In this case, the anatomical position offers limited clues as to the metastatic event order and it is equally possible that divergence into the two dominant clusters occurred in the bowel or alternatively, the bowel was independently seeded by the two clones (Fig. 5f). We detected AR splice variant transcripts (that we refer to as AR-V12 mimics, Supplementary Data 11) with a truncation at the exon 4-5 boundary, that was previously described for AR-V12/567es[26]. This was detected in 7 of 11 metastases in cluster 1 but not in any metastasis of cluster 2 ($N = 9$, Fisher's exact test, *p*-value 0.0047) suggesting expression of AR splice variants is influenced by an evolutionary-selected genomic background (Fig. 5g).

## Discussion

We here generated extensive whole genome copy number profiling on an average of 14 metastases and primary tumor samples from 10 men who died from prostate cancer after receiving abiraterone or enzalutamide. These data will serve as a resource for the community. We focused on characterizing how *AR* alterations evolve and co-exist in individual patients. We used copy number transition points rather than direct genome-wide copy number comparisons as the former can be considered the biological event that gives rise to the latter and could contain sufficient evolutionary information for accurate tracking of metastases. This strategy proved effective on FFPE (Fig. 4a) and plasma DNA samples, which introduces the opportunity to expand in future to studies with multiple temporally-separated samples.

We made a number of unique observations. Firstly, we identified a limited number of (two or three) dominant clones present at death characterized by distinct *AR* gene architectures, with at least in some cases, co-existence of both *AR* altered and gene neutral clones. This was very notably the case in CA63 with clear anatomical delineation of *AR*-gained and *AR*-neutral metastases. Interestingly, the *AR*-neutral metastases despite lower AR expression did not differ in AR activity scores compared to the gained metastases. In addition, we observed heterogeneity in *AR* alterations (either copy number alterations or break-points) in all but one patient which had a limited number ($N = 5$) of samples harvested. We also noted that there was no evidence of a predisposition of anatomical sites to a specific *AR* status: for example, in CA34, CA43 and PEA172 hepatic metastases showing a high level of *AR* gain, whilst hepatic metastases in CA63, CA76 and CA83 were clearly *AR* wild-type. Secondly, despite this intra-patient heterogeneity, *AR* alterations in clusters of metastases from the same patient were commonly in the same genomic alteration type. For example, in CA36, despite evidence of independent acquisitions, all metastases harbored one or more functionally-relevant *AR* mutations; in CA27, all metastases analyzed harbored sub-clones with the same *AR* structural variant. PEA172, CA34, CA79 and CA43 clusters had at least two different patient-unique *AR* genomic architectures achieving similar levels of *AR* copy number gain, and in CA34, we identified different frequencies of distinct break-points suggesting multiple sub-clones that had independently acquired structural change associated with similar levels of *AR* amplification. This could justify the classification of patients by *AR* gene class for therapeutic intervention, as for example is being attempted for patients harboring *AR* mutations[9,10]. However, we also showed in CA63, *AR* structurally wild-type metastases occurring alongside *AR* altered that may suggest the former metastases would be resistant to direct AR targeting despite evidence of maintaining AR transcriptional activity.

Overall, whilst we identified notable intra-patient diversity, the uniformity across clusters of metastases dominated by a single clone contextualizes previous results that might have primarily obtained samples from the same cluster[27]. Thirdly, we used autosome copy number transition points to define the evolutionary relationships of metastases and reaffirm a common clone of origin in lethal metastases and the original diagnostic biopsy. Similar approaches have been described previously[28,29] but our report describes a scalable analysis pipeline that can be implemented in whole-genome sequencing data of variable coverage and from samples of variable quality. Moreover, the patient-unique transition points common to both biopsy and autopsy sample types could be leveraged for tracking of metastatic clones by analyzing circulatory tumor DNA in plasma throughout the patient's duration of disease. Fourthly, we found in two different patients (namely CA34 and CA27) that prostate samples harvested at autopsy

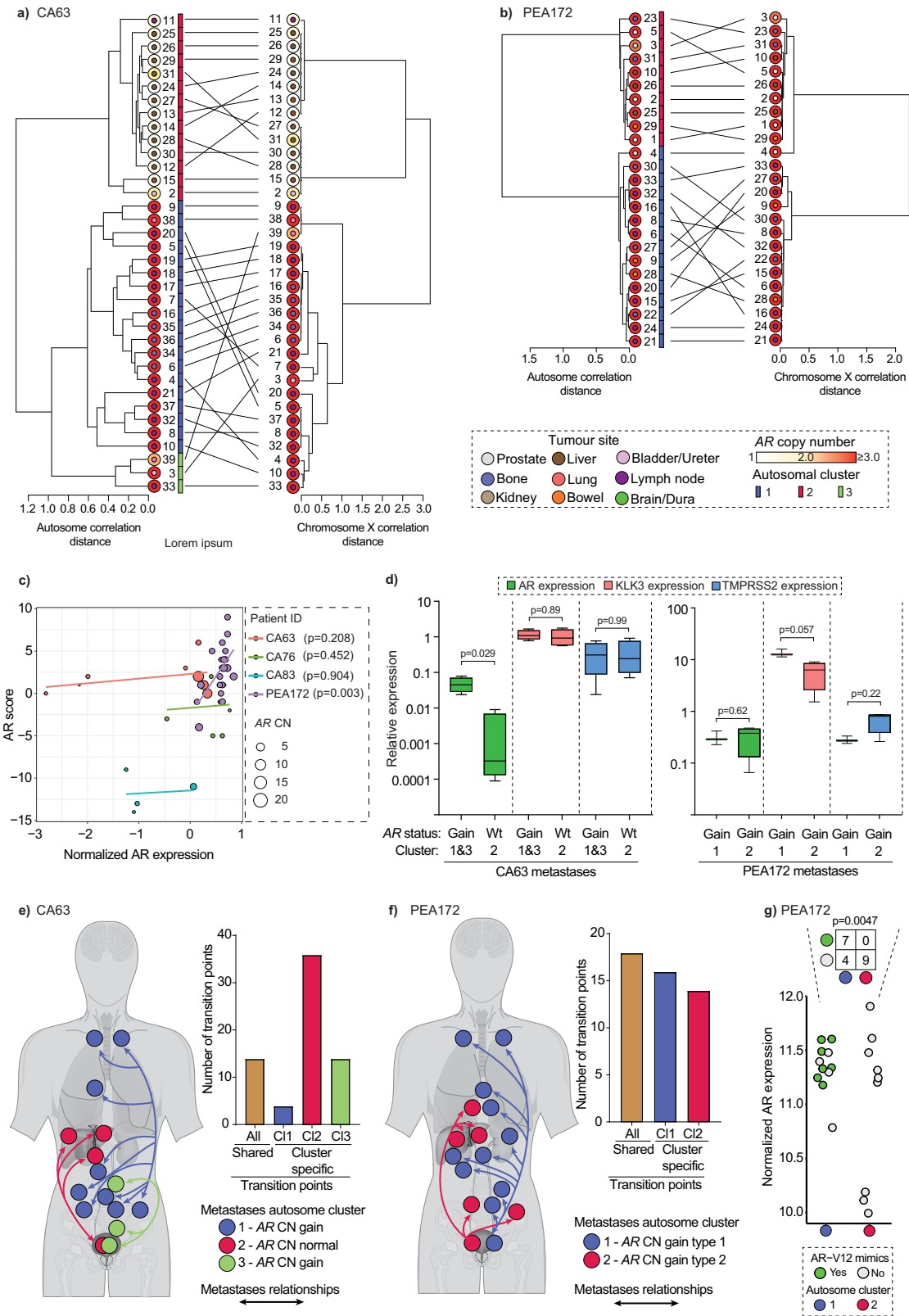

showed an absence or a very low proportion of shared transition points compared to high proportions in other metastases. This introduces the hypothesis that the prostate microenvironment exerts a different evolutionary pressure that restricts proliferation of clones that thrive in distant sites. Fifthly, using this framework, we identified a high congruence between autosomal copy number acquisition and chromosome X copy number changes, mostly clustered around the

*AR*. We hypothesize that the latter are selected at the institution of androgen signaling blockade that exerts a strong treatment-mediated selection pressure and serves as an evolutionary node through which a limited number of clones emerge, characterized by enrichment for (but not exclusively) *AR* gene structural alterations.

Our study also has some limitations. Firstly, transition points extracted from our data are biased towards events occurring at a high

**Fig. 5 | Congruence of autosomal copy number and chromosome X derived relationships suggests selection of *AR* alterations in established clones.** Tanglegrams for CA63 (**a**) and PEA172 (**b**) with autosomal SCRATCH relationship network of metastases on the left and chromosome X-based SCRATCH relationship network on the right. Gray lines between relational networks are connecting the same metastatic cores between two SCRATCH determined metastatic relationships. **c** AR scores plotted as a function of AR expression ('voom' normalized, x-axis) for CA63, CA76, CA83 and PEA172 (Supplementary Data 9). Linear regression lines are drawn for each patient with the p-values of the correlation test in legend. **d** Boxplots showing the expression of AR and two of it's regulated genes KLK3 and TMPRSS2 detected by ddPCR, on a logarithmic scale (y-axis), against the *AR* copy number status (gain or wild type) for patients CA63 ($n_{\text{gain,cluster1,3}}$ = 4 samples, $n_{\text{wt,cluster2}}$ = 4 samples) and PEA172 ($n_{\text{gain,cluster1}}$ = 3 samples, $n_{\text{gain,cluster2}}$ = 4 samples).

In each box central line represents the mean and the whiskers represent mean ± IQR * 1.5 and p-values generated from the two-tailed Mann-Whitney U test are shown on the top for each comparing pair. Cluster numbers are shown below the x-axis. **e, f** Cartoons of postulated metastatic evolutionary relationships of dominant clones for CA63 (**e**, three clusters, denoted as Cl1, Cl2 and Cl3) and PEA172 (**f**, two clusters, denoted as Cl1 and Cl2) are shown with bidirectional trajectories as reverse migration of metastatic clone(s) cannot be ruled out. Bar charts show common or cluster-specific shared copy number transition points for respective patients. **g** Normalized expression of AR gene and distribution of AR-V12 mimics in the two different SCRATCH-defined autosomal clusters (depicted with purple and red circles at the base) are shown as circles for patient PEA172 (N = 20 samples). One-sided Fisher's exact test (alternative = "less") showed a significant difference (p = 0.0047) in the distribution of AR-V12 mimics between clusters.

abundance in a tumor sample and therefore focus primarily on the "dominant" or most abundant clone. We hypothesize this clone has fitness advantages that confer resistance and we are currently unable to comment on less abundant sub-clones. Secondly, despite a large number of metastases harvested at post-mortem, we have included relatively few temporally separated samples, so our evolutionary inferences are unable to confirm "when specific clones were selected". For example, in CA27, we are unable to ascertain whether the resistant *AR*-altered clone emerged in the prostate and then seeded distant metastases or spread to the prostate after start of treatment. The former scenario would support the use of local treatment of the prostate, recently shown to improve long-term survival in low-volume, metastatic patients[30]. Similarly, although in CA63 it appears more likely that the resistant dominant clones diverged and spread from the bladder, it is also possible that one or more clones independently spread to the bladder. In the absence of multiple temporally-separated sequential samples, these patterns may be challenging to distinguish. Thirdly, a few transition points unique to the archival samples could have escaped detection in metastatic samples due to additional occurrences of structural events, later on, which could mask the common transition points while segmenting the genome with bins of similar copy number. Nonetheless, those later events in metastases could indicate the divergence of metastases leading to treatment resistance. Also, the current size of the cohort limits testing of associations between patterns of copy number clustering at death and metastatic status at presentation or distinct treatment sequences. The collection of tumors at death is pragmatic and dependent on feasibility and presence of visible tumors. We, therefore, might have over-represented clones in more accessible regions such as the liver. Forthly, as the very definition of a transition point is a function of the bin size that is selected during copy number analysis and application of segmentation algorithms, we have smoothened the data and could have grouped metastases with different breakpoints under a single copy number segment that eventually resulted in copy number changes with apparently the same transition points. Given the resolution of these data, there could be misassignment of the terminal nodes of our hierarchical clustering. Nonetheless, the dominant nodes showed a high congruence with orthogonally obtained detailed *AR* analysis.

In conclusion, we identified that a limited number of established clones with common copy number transition points are selected by treatment and are characterized by unique *AR* gene architectures. Future studies integrating temporally separated and multi-dimensional data could more closely inform on when this clonal separation occurred and in doing so identify therapeutic opportunities. Overall, our study provides further insights into the genomic evolution of prostate cancer to a lethal, drug-resistant phenotype. We hypothesize that whilst uniformity of selected *AR* alteration class supports a dependence in an individual on specific resistance escape routes, which could be disrupted by appropriately-timed targeted intervention, the co-existence of *AR*-neutral metastases suggests established clones without *AR* gene alterations can concurrently survive androgen deprivation and progress to form lethal metastases.

## Methods

### Patients and rapid warm post-mortem program
Of the ten patients involved in this study (median age 69 with a range of 56–72 years, all male), nine participated in a community-based rapid autopsy (Cancer Tissue Collection After Death, CASCADE) program described previously[31] and the remaining one patient participated in the Cancer Research UK Posthumous Evaluation of Advanced Cancer Environment (PEACE, NCT03004755) program. The inclusion criterion for this analysis was that the patient died from metastatic castration-resistant prostate cancer (mCRPC). All cases analyzed were included in this report. The number of samples to be included in the analysis was not pre-defined based on power calculations and we aimed to include the maximum number with sufficient tumor content. The CASCADE program was sponsored and conducted by the Peter MacCallum Cancer Centre, Melbourne, Victoria, Australia and approved by the Human Research Ethics Committee of the Peter MacCallum Cancer Centre, Melbourne (HREC approval numbers: CASCADE 13/122). The PEACE study was sponsored and conducted by University College London, United Kingdom and approved by the London (Dulwich) Human Research Ethics Committee (13/LO/0972). Tumor samples were obtained post-mortem, and site of biopsy was carefully annotated and photographed. Where possible, formalin-fixed paraffin-embedded (FFPE) tumor samples acquired whilst the patient was alive were retrieved. There was no randomization or blinding involved. The experimental study design is summarized in Supplementary Fig. 1. All patients provided written informed consent.

### Nucleic acid extraction
Histologically evaluated samples with sufficient tissue material were selected. DNA and RNA from fresh frozen autopsy samples were extracted using the Qiagen Allprep DNA/RNA mini kit (Qiagen) according to the manufacturer's instructions. Archival tumor blocks were retrieved retrospectively, and areas of tumor identified on Haematoxylin and Eosin slides were dissected. Germline DNA was extracted from white blood cells using QIAamp DNA Micro Kit (Qiagen).

### Library preparation and next generation sequencing
DNA libraries were prepared using the NEBNext® Ultra™ II DNA library prep kit (New England Biolabs) according to the manufacturer's protocol. In brief, 100 ng of DNA input was used for fresh frozen tumor and germline DNA while FFPE DNA inputs ranged from 5-10 ng and included an FFPE DNA repair step (New England Biolabs). DNA was sonicated on the Covaris E220 (Covaris) to a size of 150-200 bp. Sheared DNA was adapter-ligated, size selected using Agincourt AMPure beads (Beckman Coulter) and PCR amplified using unique dual index primers to reach at least 100 ng of library. Libraries were then pooled and sequenced on a NovaSeq 6000 sequencing system (Illumina) to a desired coverage using 2 × 100 paired-end sequencing

for fresh frozen and germline DNA and 2 × 50 paired end sequencing for FFPE DNA. Whole exome libraries were captured using the Roche NimbleGen SeqCap EZ Human Exome Library v3.0 kit and then the library was prepared using KAPA LTP DNA sample preparation kit from Roche and sequenced at 100 bp paired-end reads on an Illumina HiSeq 2500 sequencing system. Characterization of the *AR* coding region was achieved through targeted next generation sequencing approaches including amplicon enrichment[6] or capture-based enrichment (Integrated DNA technologies). For capture-based enrichment, a custom capture probe panel was designed including exonic regions of the genes that were previously reported to be altered in advanced prostate cancer (Supplementary Data 10) including *AR*. Spike-in probe pools for patient-specific (CA27 and CA34) breakpoints were also designed. 200 ng of whole-genome library was used as input for the capture and 10 libraries were pooled into a single capture reaction sequenced on a MiSeq sequencing system (Illumina) aiming for a 100X coverage, using 2 × 75 paired end sequencing. RNA-Seq libraries were prepared using NEBNext® Ultra™ II Directional RNA Library Prep Kit for Illumina as per manufacturer's instruction (with a slight modification to 7 cycles of PCR). Libraries were pooled and sequenced on an Illumina NovaSeq™ 6000 sequencing system (Illumina) to a desired output of 50 million reads per sample.

## Mapping of genome sequencing data
After basic quality checking using fastqc (https://www.bioinformatics. babraham.ac.uk/projects/fastqc/), reads with adapter contamination were trimmed at the 3' end using skewer[32] where the minimum permitted read length was 50 bp. Burrows Wheeler Aligner (BWA)[33] was used to map the remaining reads against human reference genome hs37d5 using default options of the bwa mem algorithm. The resulting bam files were coordinate sorted using samtools[34] and duplicated reads were removed using picard (https://broadinstitute.github.io/picard/).

## Mapping of RNA sequencing data
RNA sequencing data were mapped using STAR 2.7.9a[35] against human reference genome hs37d5 in basic two-pass mode for splice aware read alignment. Count data over gene was generated using HTSeq-count in "union" overlap solution mode utilizing a Gene Transfer Format (GTF) file from Gencode database (https://ftp.ebi.ac.uk/pub/databases/gencode/Gencode_human/release_19/gencode.v19.annotation.gtf.gz)[36].

## Tumor content estimation
In order to determine the tumor content in low-coverage samples we exploited the availability of high-coverage WGS samples for each patient. The high-coverage samples typically had high tumor content and their purity and ploidy status could be characterized with good reliability by combining information from (a) tumor/normal read-depth ratio and (b) phased B-allele frequency (BAF) at germline heterozygous SNP. Read depth counts were calculated genome-wide in bins of 100 kb using ReadCounter() function of HMMCopy v0.1.1[37]. Germline samples were genotyped with PLATYPUS v0.8.1.2[38] and phased with BEAGLE5[39,40] using the 1000 Genomes Project phase 3 reference panel[41]. Phased BAF values for tumor samples were calculated in bins of 100 K bp, using PLATYPUS v0.8.1.2[38] and in-house scripts. We focused, in particular, on large 'anchor' regions (e.g. chromosome 8p) with low read-depth ratio and with allelic imbalance (e.g. LOH regions). The read-depth levels and BAF values imposed some constraints on the possible copy numbers of the anchor regions. We determined them by finding a solution that satisfied the constraints across all the anchor regions. Finally, we ran SEQUENZA v2.1.2[42] and verified that our solution for ploidy, purity and copy number segmentation of the anchor region was among the proposed solutions.

For samples with high tumor purity, in anchor regions with allelic imbalance, the phased BAF values split into two clearly separated distributions, corresponding to the maternal and paternal chromosomes. We used this to reconstruct the long-range haplotypes in anchor regions and corrected for 'switch-errors' made by the phasing algorithm. We then calculated the haplotype BAF in low-coverage samples and determined the tumor content from it. The underlying assumption was that the copy number of the high-coverage and low-coverage samples was the same in anchor regions. This was often, but not always, the case. We therefore relied on as many anchor regions as possible (a minimum of 3) and a visual inspection for each sample. Finally, we employed the consensus between tumor contents determined by the abovementioned procedure and the ACE estimation for each sample.

## Somatic structural variant calling
Somatic structural variations in the metastatic samples were determined against matched normal (germline) sample for each patient using Delly (v-0.7.8)[43] on high-coverage whole genome or targeted sequencing data. Structural variants called on both metastatic and matched normal samples were filtered for tumor DNA contamination in the normal sample using a maximum of 0.2 ALT support. Allelic fractions of the breakpoints were first normalized by tumor content, and then further adjusted for the local chromosome number of both variant and wild-type alleles to calculate breakpoint-specific cancer cell fraction (CCF)[44].

## Determining *AR* copy number using ACE
QDNAseq/ACE R packages[12,45] were used to determine the copy number profiles from the low-coverage WGS BAM files generated from fresh-frozen autopsy samples, plasma at death and biopsy samples using a bin size of 500 kb. To determine the autosomal copy number profiles, we excluded both chromosome Y and mitochondrial DNA and the ploidy was adjusted using median bin segment value, which was the central assumption of ACE. We used a ploidy penalty of 0.5 and lower-cellularity penalty of 0.5 to fit the "squaremodel()" function of ACE as per the author's recommendation. *AR* copy number was determined, as the median copy number of the segment covering the *AR* gene, using getadjustedsegments() and analyzegenomiclocations() functions of ACE. We used the cellularity and ploidy values from our in-house approach (described in the previous section) for this calculation, where possible. We confirmed a high correlation with *AR* copy number estimated using droplet digital PCR (Supplementary Fig. 2).

## Digital droplet PCR assay
ddPCR was performed on a QX200 system (Bio-Rad) using the ddPCR Supermix for Probes (Bio-Rad) for copy number analysis and the One-Step RT ddPCR Advance kit (Bio-Rad) for gene expression analysis. Copy number assays were performed for AR (Hs04121925_cn, Life technologies), the centromeric chromosome X transcript ZXDB (Hs02220689_cn, Life Technologies) with NSUN3 (dHsaCP2506682, Bio-Rad), HCN1, AP3B1 as the reference genes. Expression assays were performed for AR (dHsaCPE5047114), KLK3 (dHsaCPE5026548) and TMPRSS2 (dHsaCPE5051496) using GAPDH (dHsaCPE5031597) and ACTB (dHsaCPE5190200) as reference transcript. PCR reactions were prepared with 2–4 ng DNA or RNA in a total volume of 22 μl and partitioned into ~20,000 droplets per sample with an Automated Droplet generator (Bio-Rad). The PCR reaction was performed and then read on a Bio-Rad QX200 droplet reader using QuantaSoft v1.3.2.0 software for either copy number or gene expression analysis.

## Somatic mutation calling and annotation
Somatic mutations were determined from whole genome, whole exome or targeted deep sequencing reads using GATK4 Mutect2[46] with default parameters. A panel of normal, in addition to matched normal

samples in each patient, from the Broad Institute (gs://gatk-best-practices/somatic-b37/Mutect2-WGS-panel-b37.vcf) was used to filter out the false positives and gnomAD vcf file (https://gnomad.broadinstitute.org/downloads), also from the Broad Institute, was used to further filter the mutation calling based on population allele frequencies of common and rare alleles. Then the somatic mutations were further filtered using FilterMutectCalls function as per the GATK best practices, including an allelic frequency of at least 0.1 and at least 5 reads supporting the variant allele. Somatic mutations with PASS designation were then annotated for their impact using dNdScv (v-0.0.1.0)[47] and the synonymous variants were filtered out in the subsequent analyses.

### Luciferase reporter assay

Prostate cancer cell line PC3 was obtained from Americal Type Culture Collection (Cat. #CRL-1435) and cultured according to the supplier's recommendations in HAM's F-12 (Gibco) supplemented with 10% FBS (Sigma). Cells were routinely checked for *Mycoplasma* (HPA cultures) and verified by fingerprinting (Eurofins). During the luciferase reporter assay, PC-3 cells were co-transfected with a PSA-ARE3-luc luciferase reporter plasmid and a Renilla luciferase vector plus an empty, *AR*-wildtype or *AR* mutant expression plasmid. Cells were seeded in white opaque 96-well plates and grown in 10% CSS-supplemented phenol red-free RPMI 1640. Cells were then treated with the 0.1 nm R1881 or 0.1 μM prednisolone, pregnenolone, progesterone or dexamethasone for 16 h. Luciferase activity was determined using Dual-Glo according to the manufacturer's instructions (Promega) and luminescence was measured on a TopCount plate reader (Perkin-Elmer).

### Determining the relationship of metastases

Consecutive genomic bins with similar normalized reads counts derived from copy number analysis were merged to form copy number segments. A revised copy number profile of each patient was determined by the copy number value of each segment at its boundary ("Transition points"). We employed SCRATCH method to determine the relationship of tumors in a patient. "Silhouette" method, as implemented in the R package "cluster"[48], was used to identify the optimal number of clusters in each SCRATCH relational network. A cluster having less than two samples, was merged with the adjacent cluster sharing the most recent common ancestor on the SCRATCH relational network.

### Clonal decomposition and validation of SCRATCH

Clonal decomposition was performed on each metastatic sample using Sclust[23], which calculated CCF from the allelic fraction of somatic mutation applying ploidy and local copy number correction. Structural variants were used to add further granularity to the analysis. Somatic mutations that belong to cluster 0 in <sample >_mclusters.txt output files were taken as the dominant population in each metastatic sample and further annotated for the impact of mutations using dNdScv[47]. Non-silent mutations were considered when an intersection of such mutations between a pair of metastatic samples from a patient were calculated. This intersection value was further normalized by the smaller total number of the non-silent clonal mutations belonging to a sample between the pair of comparisons.

### Congruence analysis

Baker's Gamma correlation coefficient[24] was used as a measure of association (similarity) between two trees of hierarchical clustering (dendrograms) using the function cor_bakers_gamma() from R package "dendextend". For calculating the Congruence Index ($I_{cong}$)[25], precompiled scripts were used from https://github.com/damiendevienne/icong. A *p*-value of 0.01 was taken as the cutoff for determining if two trees are more congruent than by chance.

### AR score

Gene-wise count data obtained from RNA sequencing were filtered for a minimum of 10 reads in at least 90% of the samples. Then the count data were quantile normalized using voom() function of Bioconductor package "Limma"[49] and z-sores were calculated for the AR-response genes ($N = 27$) described by Hieronymus and collaborators[50]. We assigned a ternary scoring system following the formula below –

AR score for each of the genes up-regulated by AR[50]:

$$AR\ score\ (up) = \begin{cases} 1 : z \geq 1 \\ 0 : -1 < z < 1 \\ -1 : z \leq -1 \end{cases} \quad (1)$$

On the other hand, for genes reported as downregulated, a converse scoring algorithm was applied. Finally, all the AR scores ($N = 20$) were aggregated to calculate the final AR score.

$$\sum \{AR\ score(up), AR\ score(down)\} \quad (2)$$

### Detecting AR splicing variant transcripts

High confidence AR fusion transcripts (AR being the 5' gene) were determined using Arriba[51] coupled with STAR aligner[52]. Detection of chimeric reads was enabled using –chimSegmentMin 10 and –chimOutType WithinBAM parameters. Fusion transcripts were reconstructed using junction specific sequence information, and open reading frames were calculated using ORFfinder (https://www.ncbi.nlm.nih.gov/orffinder/) with default settings and the largest ORF was selected.

### Statistical analysis

All quantification and statistical computing (done using R software, unless otherwise stated in the figure legend) were performed as described in the figure legends. In brief, median coverage of the WGS samples was determined using Picard (https://broadinstitute.github.io/picard/). The RNA-seq count data were quantile normalized using "voom" function in R/Bioconductor package "Limma" while calculating AR scores (to determine AR downstream transcriptional activity). To compare AR expression or AR downstream transcriptional activities among tumors with *AR* copy number gain or normal, Mann-Whitney (one-sided) test was performed. Wilcoxon (one-sided) test was performed to compare the node distances on SCRATCH relational network between clusters or organ types. In addition, Silhouette[53] analysis was used to study the separation distance between the SCRATCH-defined clusters. Congruence analysis between autosomal- and chromosome X-based SCRATCH relational networks was performed using both Baker's Gamma index[24] and Congruence Index ($I_{cong}$)[25], using a *p*-value of 0.01 as the cutoff. One-sided Fisher's exact test was performed to test the significance of the enrichment of AR-V12 mimics in one cluster in PEA172.

### Reporting summary

Further information on research design is available in the Nature Portfolio Reporting Summary linked to this article.

## Data availability

The raw transcriptomic and genomic data generated in this study are available on request on the European Genome-phenome Archive under accession number EGAS00001006598. All researchers can obtain access by submitting a project proposal to the Data Access Committees (DAC) by contacting the corresponding author (G.A.). Requests will be handled within ~8 weeks. The DAC will also determine the length of permitted access dependent on the requirements of a specific project. Processed data (minimum dataset) generated in this study have been

deposited in Zenodo database under https://doi.org/10.5281/zenodo.8125338 [https://zenodo.org/badge/latestdoi/651489188][54]. Source data are provided with this paper.

## Code availability

The codes for the figures are provided on Github: https://github.com/AMMHasan/Copy-number-architectures-define-treatment-mediated-selection-of-lethal-prostate-cancer-clones.git. A README.md file describes the utility of codes and relevant datasets. In addition, the same codes are available on Zenodo database under https://doi.org/10.5281/zenodo.8125338 [https://zenodo.org/badge/latestdoi/651489188][54].

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

## Acknowledgements

The CASCADE program was funded initially by a grant from the Peter MacCallum Cancer Foundation. The PEACE study is supported by a CRUK Accelerator Award [C416/A21999 to C.S., M.J-H.] and the National Institute for Health Research (NIHR) Biomedical Research Centre at University College London Hospital (M.L., C.S., M.J-H., G.A.); G.A. was supported by a Cancer Research UK advanced clinician scientist fel- lowship [A22744]; M.J-H. is a Cancer Research UK Career Establishment Awardee; C.S. is a Royal Society Napier Research Professor (RSRP\R \210001). His work is supported by the Francis Crick Institute that receives its core funding from Cancer Research UK (CC2041), the UK Medical Research Council (CC2041), and the Wellcome Trust (CC2041). C.S. is funded by Cancer Research UK (TRACERx (C11496/A17786), PEACE (C416/A21999) and CRUK Cancer Immunotherapy Catalyst Net- work); Cancer Research UK Lung Cancer Centre of Excellence (C11496/ A30025); the Rosetrees Trust, Butterfield and Stoneygate Trusts; NovoNordisk Foundation (ID16584); Royal Society Professorship Enhancement Award (RP/EA/180007); National Institute for Health Research (NIHR) University College London Hospitals Biomedical Research Centre; the Cancer Research UK-University College London Centre; Experimental Cancer Medicine Centre; the Breast Cancer Research Foundation (US) (BCRF-22-157); Cancer Research UK Early Detection an Diagnosis Primer Award (Grant EDDPMA-Nov21/100034); and The Mark Foundation for Cancer Research Aspire Award (Grant

21-029-ASP). CS is in receipt of an ERC Advanced Grant (PROTEUS) from the European Research Council under the European Union's Horizon 2020 research and innovation program (grant agreement no. 835297). SL was supported by a grant from the Rosetrees Trust (M892) and AKJ by a Medical Research Council Clinical Research Training Fellowship (grant no: MR/P002072/1). The analysis in this study was funded by Cancer Research UK (A22744, A26822, G.A.), the John Black Charitable Foun- dation (G.A.) and the Bob Champion Trust (G.A.). We acknowledge funding from the National Institute for Health Research (NIHR) Biome- dical Research Centre at University College London Hospital. We thank all the donors and their families who participated in the Cancer Tissue Collection After Death (CASCADE) program, along with CASCADE investigators, CASCADE Management Committee, all staff at the Vic- torian Institute of Forensic Medicine, D. Stevens, and Tobin Brothers Funerals. We also thank Miss Kamila Sychowska for creating a human body illustration for the main figures in this manuscript and Dr Karolina Nowakowska and Mr Alexander Landless for performing the reporter- luciferase assay. For the purpose of Open Access, the author has applied a CC BY public copyright licence to any Author Accepted Manuscript version arising from this submission.

## Author contributions

Conceptualization: S.S., G.A. Investigation: S.S., G.A. Sample acquisi- tion: A.P., S.W., B.T., M.L., C.N.-L., C.S., M.J.-H., S.S., G.A. Methodology: A.M.M.H., D.W., P.C., S.L., G.A. Experimentation: D.W., A.J., A.W., S.Q.W., M.P., W.D. Data analysis: A.M.M.H., P.C., D.W., S.L., M.I., S.F. Writing manuscript: A.M.M.H., P.C., D.W., S.L., G.A. Manuscript review and comments: A.M.M.H., P.C., D.W., A.J., S.Q.W., S.W., A.P., AT, B.T., E.G., S.F., O.V., M.P., M.I., W.D., A.W., M.L., PEACE consortium, C.N.-L., C.S., M.J.-H., S.L., S.S., G.A.

## Competing interests

M.J.-H. has received funding from CRUK, NIH National Cancer Institute, IASLC International Lung Cancer Foundation, Lung Cancer Research Foundation, Rosetrees Trust, UKI NETs and NIHR. M.J.-H. has consulted for, and is a member of, the Achilles Therapeutics Scientific Advisory Board and Steering Committee, has received speaker honoraria from Pfizer, Astex Pharmaceuticals, Oslo Cancer Cluster, Bristol Myers Squibb, and is co-inventor on a European patent application relating to methods to detect lung cancer (PCT/US2017/028013). C.S. acknowl- edges grants from AstraZeneca, Boehringer-Ingelheim, Bristol Myers Squibb, Pfizer, Roche-Ventana, Invitae (previously Archer Dx Inc—col- laboration in minimal residual disease sequencing technologies), Ono Pharmaceutical, and Personalis. He is Chief Investigator for the AZ MeRmaiD 1 and 2 clinical trials and is the Steering Committee Chair. He is also Co-Chief Investigator of the NHS Galleri trial funded by GRAIL and a paid member of GRAIL's Scientific Advisory Board. He receives con- sultant fees from Achilles Therapeutics (also SAB member), Bicycle Therapeutics (also a SAB member), Genentech, Medicxi, China Innova- tion Centre of Roche (CICoR) formerly Roche Innovation Centre— Shanghai, Metabomed (until July 2022), and the Sarah Cannon Research Institute C.S has received honoraria from Amgen, AstraZeneca, Bristol Myers Squibb, GlaxoSmithKline, Illumina, MSD, Novartis, Pfizer, and Roche-Ventana. C.S. has previously held stock options in Apogen Bio- technologies and GRAIL, and currently has stock options in Epic Bioscience, Bicycle Therapeutics, and has stock options and is co- founder of Achilles Therapeutics. C.S. also declares a patent application (PCT/US2017/028013) for methods to lung cancer; targeting neoanti- gens (PCT/EP2016/059401); identifying patent response to immune checkpoint blockade (PCT/EP2016/071471), determining HLA LOH (PCT/ GB2018/052004); predicting survival rates of patients with cancer (PCT/ GB2020/050221), identifying patients who respond to cancer treatment (PCT/GB2018/051912); methods for lung cancer detection (US20190106751A1). C.S. is an inventor on a European patent application (PCT/GB2017/053289) relating to assay technology to detect tumor

recurrence. This patent has been licensed to a commercial entity and under their terms of employment C.S is due a revenue share of any revenue generated from such license(s). M.L. reports grants or contracts from BMS, Shionogi, and AstraZeneca; consulting fees from BioNTech, Bicycle Therapeutics, Janssen, Merck Sorano, Pfizer, and ADC Therapeutics; honoraria from AstraZeneca and Pfizer; and support for attending meetings or travel from MSD, Janssen, and Bayer. S.S. has served on advisory boards for Bristol Myer Squibb, Merck Sharp and Dohme, Astra Zeneca, Janssen and has received institutional grant funding from Merck Sharp and Dohme, Astra Zeneca, Amgen, and Advanced Accelerator Applications (AAA), a Novartis Company, Merck Serono and Roche/Genentech (outside the submitted work). P.C., D.W., and G.A. have a patent on blood methylation markers (GB1915469.9). G.A. received personal fees, grants, and travel support from Janssen and Astellas Pharma; personal fees or travel support from Pfizer, Novartis/AAA, Bayer Healthcare Pharmaceuticals, AstraZeneca, and Sanofi-Aventis; in addition, G.A.'s former employer, The Institute of Cancer Research, receives royalty income from abiraterone and G.A. receives a share of this income through the Institute's Rewards to Discoverers Scheme. G.A. has received research funding (institutional) from Janssen, Astellas Pharma, and Novartis. All other authors declare no potential conflicts of interest.

## Additional information

[1]University College London Cancer Institute, London, UK. [2]University College London Hospitals, London, UK. [3]Peter MacCallum Cancer Centre, Melbourne, VIC, Australia. [4]The Sir Peter MacCallum Department of Oncology, University of Melbourne, Parkville, VIC, Australia. [5]Cancer Research UK Lung Cancer Centre of Excellence, University College London Cancer Institute, London, UK. [6]Cancer Evolution and Genome Instability Laboratory, The Francis Crick Institute, London, UK. [7]Department of Oncology, University College London Hospitals, London, UK. [8]Cancer Metastasis Laboratory, University College London Cancer Institute, London, UK. [41]These authors contributed equally: A. M. Mahedi Hasan, Paolo Cremaschi, Daniel Wetterskog. ✉e-mail: g.attard@ucl.ac.uk

## PEACE consortium

Charles Swanton [5,6,7], Mariam Jamal-Hanjani [5,7,8], Simone Zaccaria[1], Sonya Hessey[2], Kai-Keen Shiu[2], John Bridgewater[2], Daniel Hochhauser[1,2], Martin Forster[1,2], Siow-Ming Lee[1,2], Tanya Ahmad[1,2], Dionysis Papadatos-Pastos[1,2,9], Sam Janes[2], Peter Van Loo[10], Katey Enfield[10], Nicholas McGranahan[1,10], Ariana Huebner[1,10], Sergio Quezada[1], Stephan Beck[1], Peter Parker[10], Tariq Enver[1], Robert E. Hynds[1], David R. Pearce[1], Mary Falzon[2], Ian Proctor[2], Ron Sinclair[2], Chi-wah Lok[2], Zoe Rhodes[2], David Moore[1,2], Teresa Marafioti[2], Miriam Mitchison[2], Peter Ellery[11], Monica Sivakumar[2], Mark Linch[1], Sebastian Brandner[2], Andrew Rowan[1], Crispin Hiley[1], Selvaraju Veeriah[1], Heather Shaw[1,12], Gerhardt Attard [1,2]✉, Cristina Naceur-Lombardelli[1,5], Antonia Toncheva[1], Paulina Prymas[1], Thomas B. K. Watkins[1], Chris Bailey[10], Carlos Martinez Ruiz[1], Kevin Litchfield[1], Maise Al-Bakir[10], Nnenna Kanu[1], Sophia Ward[10], Emilia Lim[10], James Reading[1], Benny Chain[1], Blanca Trujillo [1,2], Tom Watkins[10], Melek Akay[2], Adrienne Flanagan[1], Dhruva Biswas[1], Oriol Pich[10], Michelle Dietzen[10], Clare Puttick[10], Emma Colliver[10], Alistair Magness[10], Mihaela Angelova[10], James Black[10], Olivia Lucas[10], William Hill[10], Wing-Kin Liu[1], Alexander Frankell[10], Neil Magno[1], Foteini Athanasopoulou[10], Roberto Salgado[3,13], Claudia Lee[10], Kristiana Grigoriadis[10], Othman Al-Sawaf[1], Takahiro Karasaki[1,10], Abigail Bunkum[1], Imran Noorani[10], Sarah Benafif[14], Vittorio Barbe[10], Supreet Kaur Bola[1], Osvaldas Vainauskas[1], Anna Wingate[1], Daniel Wetterskog [1,41], A. M. Mahedi Hasan [1,41], Stefano Lise[1], Gianmarco Leone[1,2], Anuradha Jayaram[1,2], Constantine Alifrangis[2], Ursula McGovern[2], Kerstin Thol[1], Samuel Gamble[1], Seng Kuong Ung[1], Teerapon Sahwangarrom[1], Claudia Peinador Marin[1], Sophia Wong[1], Piotr Pawlik[1], Jie Min Lam[2,5], Corentin Richard[10], Roberto Vendramin[10], Krijn Dijkstra[10], Jayant Rane[2], Jerome Nicod[10], Angela Dwornik[1], Kerry Bowles[2], Rija Zaidi[1], Faye Gishen[15], Paddy Stone[2], Caroline Stirling[2], Samra Turajlic[10,16,17], James Larkin[16,17], Lisa Pickering[16], Andrew Furness[16], Kate Young[16], Will Drake[18,19], Kim Edmonds[16], Nikki Hunter[16], Mary Mangwende[16], Karla Pearce[16], Lauren Grostate[16], Lewis Au[10,16], Lavinia Spain[3], Scott Shepherd[10,16], Haixi Yan[10,16], Benjamin Shum[10,16], Zayd Tippu[10,16],

Brian Hanley[10,16,17], Charlotte Spencer[10,16], Max Emmerich[10,16], Camille Gerard[10,16], Andreas Michael Schmitt[16], Lyra Del Rosario[16], Eleanor Carlyle[16], Charlotte Lewis[16], Lucy Holt[16], Analyn Lucanas[16], Molly O'Flaherty[16], Steve Hazell[16], Hardeep Mudhar[20], Christina Messiou[16,17], Arash Latifoltojar[16,17], Annika Fendler[10], Fiona Byrne[10], Husayn Pallikonda[10], Irene Lobon[10], Alexander Coulton[10], Anne-Laure Cattin[10], Daqi Deng[10], Hugang Feng[10], Andew Rowan[10], Nadia Yousaf[16], Sanjay Popat[16], Olivia Curtis[16], Charlotte Milner-Watts[16], Gordon Stamp[10], Emma Nye[10], Aida Murra[16], Justine Korteweg[16], Denise Kelly[16], Lauren Terry[16], Jennifer Biano[16], Kema Peat[16], Kayleigh Kelly[16], Charlotte Grieco[16], Mo Linh Le[16], Paolo Davide D'Arienzo[10,16], Emma Turay[16], Peter Hill[16,21], Debra Josephs[22], Sheeba Irshad[22], James Spicer[23], Ula Mahadeva[22], Anna Green[22], Ruby Stewart[22], Natasha Wright[22], Georgina Pulman[22], Ruxandra Mitu[22], Sherene Phillips-Boyd[22], Deborah Enting[22], Sarah Rudman[22], Sharmistha Ghosh[22], Eleni Karapanagiotou[22], Elias Pintus[22], Andrew Tutt[22], Sarah Howlett[22], James Brenton[24], Carlos Caldas[24,25], Rebecca Fitzgerald[25], Merche Jimenez-Linan[26], Elena Provenzano[26], Alison Cluroe[26], Anna Paterson[26], Sarah Aitken[25,26], Kieren Allinson[26], Grant Stewart[25], Ultan McDermott[26,27], Emma Beddowes[24,25], Tim Maughan[28], Olaf Ansorge[27], Peter Campbell[27], Patricia Roxburgh[29,30], Sioban Fraser[29,30], Kevin Blyth[29,30], John Le Quesne[29,30], Matthew Krebs[31], Fiona Blackhall[32], Yvonne Summers[32], Pedro Oliveira[32], Ana Ortega-Franco[32], Caroline Dive[31], Fabio Gomes[33], Mat Carter[33], Jo Dransfield[33], Anne Thomas[34], Dean Fennell[34], Jacqui Shaw[34], Claire Wilson[34], Domenic Marrone[34], Babu Naidu[35], Shobhit Baijal[35], Bruce Tanchel[35], Gerald Langman[35], Andrew Robinson[35], Martin Collard[35], Peter Cockcroft[35], Charlotte Ferris[35], Hollie Bancroft[35], Amy Kerr[35], Gary Middleton[35], Joanne Webb[35], Salma Kadiri[35], Peter Colloby[35], Bernard Olisemeke[35], Rodelaine Wilson[35], Helen Shackleford[35], Aya Osman[35], Ian Tomlinson[35], Sanjay Jogai[36], Samantha Holden[36], Tania Fernandes[36], Iain McNeish[37], Blanche Hampton[38], Mairead McKenzie[38], Allan Hackshaw[39], Abby Sharp[39], Kitty Chan[39], Laura Farrelly[39], Hayley Bridger[39], Rachel Leslie[39] & Adrian Tookman[15,40]

[9]The Princess Alexandra Hospital, Harlow, Essex, UK. [10]The Francis Crick Institute, London, UK. [11]Leeds Teaching Hospitals NHS Trust, Leeds, UK. [12]Mount Vernon Cancer Centre, Northwood, Middlesex, UK. [13]Department of Pathology, GZA-ZNA Antwerp, Antwerp, Belgium. [14]The Whittington Hospital NHS Trust, London, UK. [15]The Royal Free Hospital, London, UK. [16]The Royal Marsden Hospital, Sutton, Surrey, UK. [17]The Institute of Cancer Research, London, UK. [18]Barts Cancer Institute, London, UK. [19]Queen Mary's University of London, London, UK. [20]Sheffield Teaching Hospitals NHS Foundation Trust, Sheffield, UK. [21]Imperial College London NHS Foundation Trust, London, UK. [22]Guy's and St Thomas' NHS Foundation Trust, London, UK. [23]King's College London, London, UK. [24]CRUK Cambridge Centre, Cambridge, UK. [25]University of Cambridge, Cambridge, UK. [26]Addenbrooke's Hospital, Cambridge, UK. [27]Wellcome Sanger Institute, Hinxton, Cambridge, UK. [28]CRUK/MRC Oxford Institute for Radiation Oncology, Department of Oncology, University of Oxford, Oxford, UK. [29]Queen Elizabeth University Hospital, Glasgow, UK. [30]Beatson Institute for Cancer Research, Glasgow, UK. [31]Cancer Research UK Manchester Institute, University of Manchester, Manchester, UK. [32]Christie Hospital, Manchester, UK. [33]The Christie NHS Foundation Trust, Manchester, UK. [34]University Hospitals of Leicester, Leicester, UK. [35]Heart of England NHS Foundation Trust, Birmingham, West Midlands, UK. [36]University Hospital Southampton NHS Trust, Southampton, Hampshire, UK. [37]Imperial College London, London, UK. [38]Independent Cancer Patients' Voice, London, UK. [39]Cancer Research UK & UCL Cancer Trials Centre, University College London, London, UK. [40]Marie Curie Hospice, London, UK.

