## [Peer Review File · Nature Communications]

Copy number architectures define treatment-mediated selection of lethal prostate cancer clones.Reviewers' Comments:

Reviewer #1:

Remarks to the Author:

One of the primary mechanisms that prostate cancer (PCa) cells evoke to survive clinical treatment with antiandrogens such as enzalutamide and abiraterone is to undergo changes in the AR genomic locus leading to 'amplification' of AR signaling. The main premise of the current paper is to understand the intra-patient heterogeneity of genomic AR alterations across spatially separated lethal prostate cancer (PCa) metastases, and also use autosomal copy number changes to evaluate intra-patient metastases to inform on evolutionary relationships. Such analyses are important for understanding disease evolution and potentially the biology driving metastatic spread. A major strength of the work is detailed mapping of genomic alterations around the AR locus in multiple human PCa metastases harvested from rapid autopsy and the demonstration of 'diverse and patient-unique alterations clustering around the AR'. A major limitation of the manuscript is the overall lack of significant conceptual advances. Also, some conclusions drawn by the authors do not seem to be strongly supported by their data.

Major points:

1. It has been well established that in response to castration and antiandrogens, PCa cells would undergo extensive structural alterations around the AR genomic locus as well as at the AR enhancers (e.g., Kumar et al., Nat Med 2016; ref. 13 and 14), which would lead to increased AR signaling amplitude and castration resistance. As the authors mention, many of the AR gains in advanced CRPC are associated with resistance to abiraterone or enzalutamide. It is therefore not surprising or novel to find these amplifications in the patient samples that received more 2nd generation ARSI treatment. The samples that received less treatment actually had fewer AR alterations (CA36, CA27, and CA83)
2. It has also been reported that inter-patient metastases manifest significant heterogeneity in genomic alterations including structural re-arrangements around the AR locus; however, there was generally a lack of intra-patient heterogeneity across metastatic sites (Kumar et al., Nat Med 2016; ref. 18). To a certain degree, the intra-patient heterogeneity in AR alterations (mutations, copy number, breakpoints etc) across different metastases reported in the current study is interesting but not necessarily surprising.
3. Moreover, the important conceptual conclusion drawn by the authors using autosome copy number transition points, i.e., metastases and the original diagnostic biopsy have 'a common clone of origin', has been known for a long while.
4. In Figure 3e, it was stated that "tumor sequencing reads with a wild-type allele at this position suggests this mutation occurred after metastatic seeding, despite its presence in every metastasis, and emerged independently in the lineages with D891N or without". This seems to be implying that AR mutations are occurring independently in each metastatic site, rather than from a clonal expansion process. There is not enough data to support this conclusion. The presence of WT alleles could simply be the result of contaminating normal tissue surrounding the tumor tissue. The authors' tumor content prediction suggests that every tumor specimen would have some normal/benign tissues (even infiltrating lymphocytes) which would result in some WT alleles to be present. It is difficult to determine from the presented data which events arose independently or from clonal dissemination.
5. Results Figure 3f are a bit difficult to interpret and are inconsistent with the authors' conclusion that 'We found that acquiring a T878A change at a D891N altered allele'.
6. Correlations between AR expression and AR score in CA63, CA76, and CA83 are discussed as being weak but positive (Figure 5c). A close examination of the data suggests that these correlations may not be relevant as they're pretty much flat. What is the significance (P value) of these correlations? These correlations may be weak since they are made relative to AR mRNA and not protein expression.

Minor points:

1. It is unclear which samples are archival versus which samples are collected post-mortem.
2. In figure S3, the legend is titled AR CN for the scale. Only a small portion of the X chromosome is related to AR, yet the scale is used across the chromosomal binning. The scale should be more

generally titled CN, not AR CN.

3. In figure S6, the authors claim that the SCRATCH relationship network assigned plasma as having the highest similarity to liver metastases. Is there some statistical analysis that can back up this claim? It seems to be quite variable across most patient samples. In one patient the liver mets are split between the two main clusters and one of the clusters contains the plasma sample. So does the plasma sample only match half of the liver mets?

4. In Discussion, the authors mentioned that '..... there was no evidence of a predisposition of specific anatomical sites to a specific AR status' (page 13, line 350). However, the AR 'pC687Y was detected solely in the brain metastases' (page 6, bottom line).

Reviewer #2:

Remarks to the Author:

This manuscript by Hasan et al. performed an in-depth copy number profiling of 167 metastatic regions from 10 men who died of prostate cancer. From the analysis, the authors identified several dominant AR clones at death and heterogeneity of AR alterations. Furthermore, the authors developed the algorithm SCRATCH to study the relationship of tumors from different sites using the copy numbers at transition points. This is a well-structured and organized study. The figures are generally clearly presented and described throughout the manuscript, especially Figure 1b which clearly illustrated the different metastatic sites and their corresponding AR copy numbers. And the hypothesis that the tumor microenvironment could selectively allow certain clones to survive is particularly interesting. This study could provide a valuable source for the field of understanding the impact of AR copy numbers in the context of mCRPC.

1. In the metastatic site distribution, there is a vast difference in samples in different sites. For example, CA35, CA43, CA63, CA34, and CA83 do not have any samples from the brain. And only a subset of patients has lung samples. It is curious how this bias towards certain sites or certain patients could affect the conclusions of this study considering a great portion of the analysis is focused on comparing AR copy numbers from different metastatic sites.

2. It is very interesting that the patient with the highest density of AR breakpoints also has TMPRSS2-ERG fusion events. Are there any other patients in this cohort also having ETS rearrangements? If so, should the authors investigate the AR-associated breakpoints by not only sites but also the status of ETS rearrangement?

3. The authors also investigated the onset of AR mutations. When calling the mutations using Mutect2, did the authors filter the Mutect2 calls using the orientation bias model in GATK? Since there are FFPE samples in this study, the authors might want to take into consideration the false positives due to FFPE. And what other filter/curation did the authors use to identify the AR-associated mutations? Since the allele frequencies of T878A are generally very high in CA36, how do these mutations compare to the other patients in this cohort (are there also T878A mutations?) or previous studies?

4. In Figure 5d, it is shown that within CA63, samples without AR amplification would have lower AR expression. But it is not the case for CA76 where the AR expressions are almost at the same level as CA63 samples with AR amplification. Could the authors comment on that? And why select TMPRSS2 for expression analysis? If the authors are interested in TMPRSS2-ERG fusion status, should it be more suitable to check ERG expression? Or other PCa luminal markers such as FOLH1 or NKX3-1?

5. Did the authors attempt to validate the conclusions in another mCRPC cohort as a positive control? Previous studies indicated that the general mutational profiles for mCRPC patients differ by metastatic sites therefore it is extremely intriguing that there exist some common clones in the copy number profile. Furthermore, do the copy number changes co-occur with any rearrangement events?

Reviewer #1:

One of the primary mechanisms that prostate cancer (PCa) cells evoke to survive clinical treatment with antiandrogens such as enzalutamide and abiraterone is to undergo changes in the AR genomic locus leading to ‘amplification’ of AR signaling. The main premise of the current paper is to understand the intra-patient heterogeneity of genomic AR alterations across spatially separated lethal prostate cancer (PCa) metastases, and also use autosomal copy number changes to evaluate intra-patient metastases to inform on evolutionary relationships. Such analyses are important for understanding disease evolution and potentially the biology driving metastatic spread. A major strength of the work is detailed mapping of genomic alterations around the AR locus in multiple human PCa metastases harvested from rapid autopsy and the demonstration of ‘diverse and patient-unique alterations clustering around the AR’. A major limitation of the manuscript is the overall lack of significant conceptual advances. Also, some conclusions drawn by the authors do not seem to be strongly supported by their data.

We thank Reviewer #1 for their positive comments. We address the reviewer’s concern regarding conclusions “not strongly supported by the data” in our specific replies. We believe that these were mostly related to our text not being sufficiently clear, so we have amended it accordingly. For clarity, we list some of the conceptual advances and novel findings included in our report -

1. We used a bespoke novel approach (SCRATCH) adapted for low pass whole genome sequencing of tumors of variable quality (FFPE, fresh frozen) and identified Transition Points (as representatives of the start of genomic events) in temporally-separated diagnostic biopsies and post-mortem sampling. To the best of our knowledge, this is the first study to this type of evolutionary-defined clusters of metastases. As our approach works on samples of variable quality, we were also able to confirm a common lethal clone in both diagnostic and post-mortem tumors; providing proof of concept for temporal tracking of tumor clones, which can be implemented on fresh frozen or formalin fixed paraffin embedded samples as well as plasma DNA.
2. We included a large numbers of tumors per patient (35 in CA63 and, 26 each in CA83 and PEA172) that probably represents the largest number of intra-patient sampling after modern-day therapies. This has allowed us to conclude that a limited number (2 to 3) of dominant clones emerge under treatment-mediated selection pressure (irrespective of the number of metastases analysed per patient).
3. We identified a high congruence between autosomal copy number acquisition (defined by SCRATCH) and chromosome X copy number changes (mostly involving or close to the *AR*), based on which we hypothesized that the later events were selected by treatment mediated strong selection pressure and served as an evolutionary node (bottleneck) for a limited number of clones (characterized by *AR* gene architecture) to pass through. This discovery, alongside co-existence of clusters of metastases with or without *AR* alterations, is a novel and intriguing discovery.
4. We present the evolutionary landscape of *AR* alterations in a novel way, including using mutations at p.T878A and p.D891N to identify the acquisition of mutations via at least 3 separate events, skyline plots of *AR* copy number change to display structural differences that

putatively arise from distinct treatment pressures (**Figure 2**) and the description of several novel *AR* structural rearrangements.

5. We show evidence of maintained AR signalling in *AR* wild-type metastases that have evolved alongside AR gained metastases and we also show expression of the AR splice variant (AR-V12 mimic) in an evolution-selected genomic background (only one genomically-defined cluster of metastases had splice variants, **Figure 5g**).

Major points:

1. It has been well established that in response to castration and antiandrogens, PCa cells would undergo extensive structural alterations around the AR genomic locus as well as at the AR enhancers (e.g., Kumar et al., Nat Med 2016; ref. 13 and 14), which would lead to increased AR signaling amplitude and castration resistance. As the authors mention, many of the AR gains in advanced CRPC are associated with resistance to abiraterone or enzalutamide. It is therefore not surprising or novel to find these amplifications in the patient samples that received more 2nd generation ARSI treatment. The samples that received less treatment actually had fewer AR alterations (CA36, CA27, and CA83)

We agree with Reviewer #1 that it is not surprising that the *AR* has undergone extensive structural alterations in our cohort, who had invariably received current standard-of-care, potent second-generation ARSI treatment. Nonetheless, this is an important result for our story: we present this in our first results section to set the scene (page 11, line 290-293):

“Overall, we effectively confirmed diverse and patient-unique alterations clustering around the AR and its enhancer in metastases from every patient, supporting the potent selective pressures for AR aberrant clones in men receiving hormonal therapies.”

We agree with Reviewer #1 that there is an intriguing association between the number of breakpoints detected at the *AR* locus (around gene and centromeric enhancer) and the duration of second-generation ARSI treatment. We have added this hypothesis-generating observation to the main text of the manuscript (page 10-11 line 262-266):

“Intriguingly, we found that patients (CA27, CA36 and CA83) with the shortest exposure (2,3, 2 months respectively compared to 4, 6, 10, 11, 12, 22 and 32 months for the rest of the cohort) to first-line second-generation ARSI (abiraterone or enzalutamide, **Table S1**) had the fewest break-points detected at the *AR* locus (1.67 versus 6.43 break-points detected per sample per patient)”

2. It has also been reported that inter-patient metastases manifest significant heterogeneity in genomic alterations including structural re-arrangements around the AR locus; however, there was generally a lack of intra-patient heterogeneity across metastatic sites (Kumar et al., Nat Med 2016; ref. 18). To a certain degree, the intra-patient heterogeneity in AR alterations (mutations, copy number, breakpoints etc) across different metastases reported in the current study is interesting but not necessarily surprising.

We agree with the reviewer, that the publication by Kumar *et al.*¹ is the landmark study that defines the current field. Quoting the Kumar report:

“AR genomic aberrations were generally concordant within individuals: in eight men, all tumors were consistent in having no AR aberrations, and for 24 men, all tumors were found to have either AR copy gain or mutation. In nine men, there was discordance, with one or more tumors exhibiting an AR copy gain or mutation and one or more tumors showing no aberration. We identified one man with evidence of convergent evolution, in whom several metastases had AR mutation and other metastases had AR amplification, each potentially a contributor to treatment resistance.”

In fact, we started our study soon after publication of the Kumar *et. al.* ¹, aiming to interrogate in greater the intra-patient heterogeneity in AR status that they described. Critically we provide a major paradigm advance: metastases group into limited numbers of clusters that share the same AR alteration, with distinct clusters of metastases having different chromosome X or AR genomic architectures. This supports convergent evolution in some cases (*i.e.* PEA172, CA63 and CA79) where distinct clusters acquired different alterations in AR but our report also shows that alternative paths exist for treatment escape (with CA63 having a cluster not harboring any AR copy number alteration versus two clusters with AR copy number gains).

3. Moreover, the important conceptual conclusion drawn by the authors using autosome copy number transition points, i.e., metastases and the original diagnostic biopsy have ‘a common clone of origin’, has been known for a long while.

We agree with the reviewer, and had stated in our results section (page 14, line 360), “This reaffirms a common clone of origin in metastases at death”.

We had also further addressed in our discussion section (page 20, line 525-528): “Thirdly, we used autosome copy number transition points to define the evolutionary relationships of metastases and reaffirm a common clone of origin in lethal metastases and the original diagnostic biopsy”.

However, to the best of our knowledge, our study is the first to show a monoclonal origin in temporally-separated samples – *i.e.*, tumor obtained from the same patient at diagnosis and post-mortem, separated by ~3 - 10 years.

4. In Figure 3e, it was stated that “tumor sequencing reads with a wild-type allele at this position suggests this mutation occurred after metastatic seeding, despite its presence in every metastasis, and emerged independently in the lineages with D891N or without”. This seems to be implying that AR mutations are occurring independently in each metastatic site, rather than from a clonal expansion process. There is not enough data to support this conclusion. The presence of WT alleles could simply be the result of contaminating normal tissue surrounding the tumor tissue. The authors’ tumor content prediction suggests that every tumor specimen would have some normal/benign tissues (even infiltrating lymphocytes) which would result in some WT alleles to be present. It is difficult to determine from the presented data which events arose independently or from clonal dissemination.

We agree with the reviewer that the 99 wild-type sequencing reads in CA36_11 (**Figure 3e**) are unlikely to be cancer in origin. Specifically, we referred (in the main text) to the reads that are wild-type for p.T878 but mutant for p.D891 (393 reads) and vice versa (*i.e.* tumour derived). We have updated the text on page 12-13, line 312-328 to clarify the results:

“Specifically we identified 364 sequencing reads harboring mutations at both positions X:66943591 (that codes for p.D891N) and X:66943552 (coding for p.T878A) in the same sample with reads that were mutant at one of these positions but wild-type at the other (6 reads with wild-type p.D891 but mutant p.T878A, and 393 reads with mutant p.D891N but wild-type p.T878, **Figure 3e**). Although this means that cells with one mutation acquired the other, we cannot distinguish the order of events. To further investigate the biological pressures for the selection of these two AR mutations, we used a reporter luciferase construct in prostate cancer cells transfected with AR wild-type or a combination of the detected mutations and treated with a range of ligands, including progesterone and prednisone previously suggested as contributing to resistance against abiraterone^{14,36} (**Figure 3f**). We found that acquiring AR p.D891N alone did not confer significant increased activation by a range of ligands compared to AR wildtype. Overall, this suggests that despite being present in all metastases analysed, the p.T878A mutation occurred after metastatic seeding and in this liver metastasis emerged independently in a lineage with p.D891N mutation. Based on our functional assays, it may be more likely that pD891N mutant cells acquired a pT878A than vice versa. We deemed the alternative explanation of reverting a mutation to its wild-type as unlikely although we are unable to exclude this possibility.”

5. Results Figure 3f are a bit difficult to interpret and are inconsistent with the authors’ conclusion that ‘We found that acquiring a T878A change at a D891N altered allele’.

We have expanded this result to make this clearer on page 12-13, line 316-326:

“Although this means that cells with one mutation acquired the other, we cannot distinguish the order of events. To further investigate the biological pressures for the selection of these two AR mutations, we used a reporter luciferase construct in prostate cancer cells transfected with AR wild-type or a combination of the detected mutations and treated with a range of ligands, including progesterone and prednisone previously suggested as contributing to resistance against abiraterone^{14,36} (**Figure 3f**). We found that acquiring AR p.D891N alone did not confer significant increased activation by a range of ligands compared to AR wildtype. Overall, this suggests that despite being present in all metastases analysed, the p.T878A mutation occurred after metastatic seeding and in this liver metastasis emerged independently in a lineage with p.D891N mutation. Based on our functional assays, it may be more likely that pD891N mutant cells acquired a pT878A than vice versa.”

6. Correlations between AR expression and AR score in CA63, CA76, and CA83 are discussed as being weak but positive (Figure 5c). A close examination of the data suggests that these correlations may not be relevant as they’re pretty much flat. What is the significance (P value) of these correlations? These correlations may be weak since they are made relative to AR mRNA and not protein expression.

We have expanded our results (page 17, line 437 – 452) to clarify these findings:

“We compared AR expression data from RNA sequencing on metastases showing intra-patient differences in AR status (N=39 from CA63, CA76, CA83 and PEA172) with AR transcriptional activity (AR score) (**Figure 5c**). We found a weak but positive correlation between AR copy number gain and normalized AR expression (*p-value* 0.009) and between AR copy number gain and AR score (*p-value*

0.03, **Figure S7**) but when analyzing tumors grouped by patient, we observed a more complex picture. There was no difference in AR score across tumors with low and high levels of AR expression in three patients (CA63, CA76 and CA83) whilst in PEA172, characterized by high AR copy numbers and consequently relatively high AR expression in all tumors, AR expression and score were correlated (*p-value* 0.003) (**Figure 5c, d and S8**). We confirmed consistently high levels of the AR-response genes KLK3 and TMPRSS2 in tumors from CA63, CA76 and CA83 with and without AR gain (and respectively high and low AR expression) using ddPCR (**Figure 5d and S8**). We also found in PEA127, in keeping with high AR copy number levels, high AR, KLK3 and TMPRSS2 expression across all tumors, with higher KLK3 levels in cluster 1 versus 2, confirming differential AR activity between metastases from the two AR-gained but structurally distinct clusters.”

We are not able to perform quantitative AR protein assessment due to the absence of FFPE slides on these samples. However, we have clarified the key result relates to individual patient complexity with matching AR score and AR expression levels (measured both by RNASeq and ddPCR).

Minor points:

1. It is unclear which samples are archival versus which samples are collected post-mortem.

Archival samples are included solely in **Figure 4a** as stated in text (page14, line 358-360):

“we uniquely observed that transition points in archival FFPE samples from an individual were shared with all metastases harvested post-mortem from the same patient (putatively patient-common or truncal transition points) (**Figure 4a and S5**).”

To ensure clarity, we have added a reference to **Figure 4a** in our discussion as well. Moreover, we have added a column in **Table S2** that annotates whether samples are FFPE or fresh frozen (FF).

2. In figure S3, the legend is titled AR CN for the scale. Only a small portion of the X chromosome is related to AR, yet the scale is used across the chromosomal binning. The scale should be more generally titled CN, not AR CN.

We agree and have amended **Figure S3**.

3. In figure S6, the authors claim that the SCRATCH relationship network assigned plasma as having the highest similarity to liver metastases. Is there some statistical analysis that can back up this claim? It seems to be quite variable across most patient samples. In one patient the liver mets are split between the two main clusters and one of the clusters contains the plasma sample. So does the plasma sample only match half of the liver mets?

We were intrigued that plasma was invariably assigned with liver metastases but we agree with the reviewer that this result is hard to prove with this number of patients. We have therefore removed this sentence from our text (page 16, line 434). We retained **Figure S6** that displays the clusters output from the SCRATCH network. Readers can draw their own conclusions from the observation of plasma being assigned to the same cluster as liver metastases in these data and we and other may prove this in future larger studies.

4. In Discussion, the authors mentioned that ‘..... there was no evidence of a predisposition of specific anatomical sites to a specific AR status’ (page 13, line 350). However, the AR ‘pC687Y was detected solely in the brain metastases’ (page 6, bottom line).

We included 13 metastases from brain or dura from four patients. Only 2 of 3 metastases from patient CA36 harbor the mutation p.C687Y. This mutation occurred independently in these two metastases, but we do not consider this as a pre-disposition to this site. For example, we also showed evidence of independent acquisition of p.D891N variant in liver metastases from this patient and do not believe this should be considered as evidence that p.D891N alteration is predisposed to occur in liver. Specifically, in this discussion point we were emphasising that liver metastases (that can clinically be associated with neuro-endocrine transformed disease) could as commonly harbour *AR* structural rearrangements. We believe that given the number of sites we sampled and the extensive analysis of *AR*, we are justified in saying that there is no evidence that specific sites show a pre-disposition for specific *AR* alterations.

Reviewer #2:

This manuscript by Hasan et al. performed an in-depth copy number profiling of 167 metastatic regions from 10 men who died of prostate cancer. From the analysis, the authors identified several dominant AR clones at death and heterogeneity of AR alterations. Furthermore, the authors developed the algorithm SCRATCH to study the relationship of tumors from different sites using the copy numbers at transition points. This is a well-structured and organized study. The figures are generally clearly presented and described throughout the manuscript, especially Figure 1b which clearly illustrated the different metastatic sites and their corresponding AR copy numbers. And the hypothesis that the tumor microenvironment could selectively allow certain clones to survive is particularly interesting. This study could provide a valuable source for the field of understanding the impact of AR copy numbers in the context of mCRPC.

We thank the reviewer for the positive comments.

1. In the metastatic site distribution, there is a vast difference in samples in different sites. For example, CA35, CA43, CA63, CA34, and CA83 do not have any samples from the brain. And only a subset of patients has lung samples. It is curious how this bias towards certain sites or certain patients could affect the conclusions of this study considering a great portion of the analysis is focused on comparing AR copy numbers from different metastatic sites.

The reviewer is probably right that every tissue analysis study is biased towards more accessible metastases that can be sampled. By including tumor samples acquired post-mortem, our study removes some of these biases or barriers (given less accessible metastases can be included *e.g.* brain, lung). In fact, we present data on metastases from 11 different organs (**Figure 1**). Importantly, the split of metastatic sites in our study is representative of clinical disease with bone and liver samples, which are by far the most common sites of metastases in prostate cancer patients, accounting for most

samples analysed. Overall, given the wide range of sites sampled we see no evidence that our conclusions are confounded by certain sites but we include in our discussion (page 21, line 565-567):

“The collection of tumors at death is pragmatic and dependent on feasibility and presence of visible tumors. We, therefore, might have over-represented clones in more accessible regions such as the liver.”

2. It is very interesting that the patient with the highest density of AR breakpoints also has TMPRSS2-ERG fusion events. Are there any other patients in this cohort also having ETS rearrangements? If so, should the authors investigate the AR-associated breakpoints by not only sites but also the status of ETS rearrangement?

We explored the reviewer’s hypothesis but found CA35 and CA36 that are also TMPRSS2-ERG fusion positive have very few *AR* breakpoints. We have now provided TMPRSS2-ERG status (the most common structural rearrangement in prostate cancer) for all 10 patients in **Table S12** (only for samples with high coverage sequence reads). This will increase the value of our dataset as a resource for the community.

3. The authors also investigated the onset of AR mutations. When calling the mutations using Mutect2, did the authors filter the Mutect2 calls using the orientation bias model in GATK? Since there are FFPE samples in this study, the authors might want to take into consideration the false positives due to FFPE. And what other filter/curation did the authors use to identify the AR-associated mutations?

We did not call SNVs on FFPE samples (for the reasons expressed by the reviewer).

We have added the following filters that were used to the **Methods** section in **Supplementary materials** (page 5, line 119):

1. Default mutect2 filter for mapping quality and strand bias.
2. Excluding variants lacking a PASS designation.
3. An allelic fraction of 0.1 or above.
4. A minimum of 5 reads for variant allele.

4. Since the allele frequencies of T878A are generally very high in CA36, how do these mutations compare to the other patients in this cohort (are there also T878A mutations?) or previous studies?

Although we included samples from multiple patients in our sequencing workflow, we only detected T878A in metastases from CA36 (*i.e.* one out of 10 patients). This is a prevalence that is consistent with prior larger studies (but generally with single tumor samples from every patient) that describe p.T878A in ~3-15% cases²⁻⁶.

5. In Figure 5d, it is shown that within CA63, samples without AR amplification would have lower AR expression. But it is not the case for CA76 where the AR expressions are almost at the same level as CA63 samples with AR amplification. Could the authors comment on that? And why select *TMPRSS2* for expression analysis? If the authors are interested in *TMPRSS2*-*ERG* fusion status, should it be more suitable to check *ERG* expression? Or other PCa luminal markers such as *FOLH1* or *NKX3-1*?

To reduce outlier results in individual samples, we have improved our presentation of these data with the following figure (Figure 5c and d), retaining cases where we had sufficient samples to perform statistical analysis on:

Data on other patients has been moved to the supplementary Figure S8.

We used *TMPRSS2* and *KLK3* as two of the more AR-responsive genes as described by us and others previously⁷. *ERG* is AR regulated due to a *TMPRSS2*-*ERG* gene fusion in 4 of the patients (Table S12) in this cohort and therefore not as useful for including in these analyses. In addition, *FOLH1* and *NKX3.1* are not as AR responsive so we are not convinced they would be useful for these analyses.

6. Did the authors attempt to validate the conclusions in another mCRPC cohort as a positive control?

Our findings are consistent across all ten patients. There is currently no independent validation cohort available for testing, but we argue this is not necessary given the consistency across all cases.

7. Previous studies indicated that the general mutational profiles for mCRPC patients differ by metastatic sites therefore it is extremely intriguing that there exist some common clones in the copy number profile.

The common copy number profiles are “truncal” and present prior to evolutionary branching of our metastases (SCRATCH) clusters. This is consistent with “truncal” mutation profiles in studies of other cancer types. The reviewer is probably alluding to “private” mutations that differ by metastatic site. Similarly, we identified transition points that differ by metastatic sites and underly the clustering we report (Figure 4d and Figure S6).

8. Furthermore, do the copy number changes co-occur with any rearrangement events?

We found no evidence of frequent co-occurrence of copy number changes and rearrangement events. This makes sense as several types of structural rearrangements (detected by breakpoints in sequence reads in our analysis) are copy number neutral.

References used in this discussion:

- 1 Kumar, A. *et al.* Substantial interindividual and limited intraindividual genomic diversity among tumors from men with metastatic prostate cancer. *Nat Med* **22**, 369-378, doi:10.1038/nm.4053 (2016).
- 2 Romanel, A. *et al.* Plasma AR and abiraterone-resistant prostate cancer. *Sci Transl Med* **7**, 312re310, doi:10.1126/scitranslmed.aac9511 (2015).
- 3 Robinson, D. *et al.* Integrative clinical genomics of advanced prostate cancer. *Cell* **161**, 1215-1228, doi:10.1016/j.cell.2015.05.001 (2015).
- 4 Lallous, N. *et al.* Functional analysis of androgen receptor mutations that confer anti-androgen resistance identified in circulating cell-free DNA from prostate cancer patients. *Genome Biol* **17**, 10, doi:10.1186/s13059-015-0864-1 (2016).
- 5 Ledet, E. M. *et al.* Comprehensive Analysis of AR Alterations in Circulating Tumor DNA from Patients with Advanced Prostate Cancer. *The Oncologist* **25**, 327-333, doi:10.1634/theoncologist.2019-0115 (2019).
- 6 Quigley, D. A. *et al.* Genomic Hallmarks and Structural Variation in Metastatic Prostate Cancer. *Cell* **174**, 758-769.e759, doi:10.1016/j.cell.2018.06.039 (2018).
- 7 Carreira, S. *et al.* Tumor clone dynamics in lethal prostate cancer. *Science Translational Medicine* **6**, doi:10.1126/scitranslmed.3009448 (2014).

Reviewers' Comments:

Reviewer #1:

Remarks to the Author:

The authors made conscientious efforts to revise the manuscript which has improved significantly with respect to clarity and readability. Also to the authors' credit, potential limitations of the study were thoroughly discussed. The results presented herein could represent a valuable resource for scientists in the field in future studies of attempting to understand and track the clonal evolution of metastatic clones in mCRPC patients treated with the next generation of ARSIs. There are just some grantsmanship issues and inconsistency in data presentation.

1. Figure S1: The actual content presented in this figure seems to be truncated (cut off).
2. Figure 2: The legend of Figure 2 (h & i) "Inset used to show" should be Figure 2 (e).
3. Figure 3b: Both the Text (page 11, lines 275 – 278) and the legend talked about 'break point associated with TMPRSS-ERG fusion' but this was NOT shown in the actual Figure 3b. Also, in Figure 3b legend (page 28, line 777), CA34_4 should be CA34_3.
4. Figure S5: The legend says "Circles representing proportion of" but there are no circles. Also, the legend says (n=7 patients) but Patient IDs show 8 patients.
5. Figure S6: Data for patient PEA172 were not shown.
6. Page 16 (line 426): CA34 should be PEA172.
7. Page 18, lines 461, 480 and 485: Figure 5c, Figure 5d and Figure 5e should be Figure 5e, Figure 5f and Figure 5g, respectively.

Reviewer #2:

Remarks to the Author:

The authors have adequately addressed all the previous comments.

Here we have responded to the reviewers' last comments regarding the merit of our study and also to the minor points that they have raised. Our responses below are provided in normal fonts, while the comments from the reviewers are in bold. In addition, the page numbers mentioned in this reply correspond to the word documents **Manuscript_NCOMMS-22-19944B_track_changed** or the **supplementary information** (as mentioned below):

Reviewer #1 (Remarks to the Author):

The authors made conscientious efforts to revise the manuscript which has improved significantly with respect to clarity and readability. Also to the authors' credit, potential limitations of the study were thoroughly discussed. The results presented herein could represent a valuable resource for scientists in the field in future studies of attempting to understand and track the clonal evolution of metastatic clones in mCRPC patients treated with the next generation of ARSIs.

We sincerely appreciate the comments of Reviewer 1 and the remarks on the contribution of this study to the wider science community.

There are just some grantsmanship issues and inconsistency in data presentation.

1. Figure S1: The actual content presented in this figure seems to be truncated (cut off).

We have reviewed **Figure S1** and a completed figure is reposted in the Supplementary materials.

2. Figure 2: The legend of Figure 2 (h & i) "Inset used to show" should be Figure 2 (e).

We have rectified the error and verified the accurate label in the legend of **Figure 2**.

3. Figure 3b: Both the Text (page 11, lines 275 – 278) and the legend talked about 'break point associated with TMRSS-ERG fusion' but this was NOT shown in the actual Figure 3b. Also, in Figure 3b legend (page 28, line 777), CA34_4 should be CA34_3.

We really appreciate the reviewer's observation and we have rectified the mistake. Now we have incorporated, into **Figure 3b**, a column of pie charts depicting the cancer cell fraction of TMRSS-ERG fusion breakpoint in 11 samples of patient CA34. In addition, the sample identity of CA34_3 has also been corrected in the legend of Figure 3b (line 838).

4. Figure S5: The legend says "Circles representing proportion of" but there are no circles. Also, the legend says (n=7 patients) but Patient IDs show 8 patients.

We have now redrawn **Figure S5** with the corrected patient number in the figure legend. While **Figure 4a** depicts the comparisons of all archival (FFPE) samples (24 samples from eight patients) to their corresponding metastases, **Figure S5** depicts the same comparisons involving only the diagnostic prostate FFPE samples (18 samples from seven patients).

5. Figure S6: Data for patient PEA172 were not shown.

Now the SCRATCH relationship of the metastases in patient PEA172 has been added in **Figure S6** (line 248 of Supplementary Information).

6. Page 16 (line 426): CA34 should be PEA172.

Patient identity has been corrected accordingly (line 282).

7. Page 18, lines 461, 480 and 485: Figure 5c, Figure 5d and Figure 5e should be Figure 5e, Figure 5f and

Figure 5g, respectively.

The correct references of the corresponding figures in the text have been updated accordingly (lines 316, 335 and 340, respectively).

Reviewer #2 (Remarks to the Author):

The authors have adequately addressed all the previous comments.

We sincerely appreciate the valuable comment provided by Reviewer 2, and we are pleased to note that we have successfully addressed his/her previous comments.